# Equivariant Neural Networks for General Linear Symmetries on Lie Algebras

Chankyo Kim [* 1]   Sicheng Zhao [* 1]   Minghan Zhu [2]   Tzu-Yuan Lin [3]   Maani Ghaffari [1]

## Abstract

Many scientific and geometric problems exhibit general linear symmetries, yet most equivariant neural networks are built for compact groups or simple vector features, limiting their reuse on matrix-valued data such as covariances, inertias, or shape tensors. We introduce **Reductive Lie Neurons (ReLNs)**, an exactly $\mathrm{GL}(n)$-equivariant architecture that natively supports matrix-valued and Lie-algebraic features. ReLNs resolve a central stability issue for reductive Lie algebras by introducing a non-degenerate adjoint (conjugation)-invariant bilinear form, enabling principled nonlinear interactions and invariant feature construction in a single architecture that *transfers across subgroups without redesign*. We demonstrate ReLNs on algebraic tasks with $\mathfrak{sl}(3)$ and $\mathfrak{sp}(4)$ symmetries, Lorentz-equivariant particle physics, uncertainty-aware drone state estimation via joint velocity–covariance processing, learning from 3D Gaussian-splat representations, and EMLP double-pendulum benchmark spanning multiple symmetry groups. ReLNs consistently match or outperform strong equivariant and self-supervised baselines while using substantially fewer parameters and compute, improving the accuracy–efficiency trade-off and providing a practical, reusable backbone for learning with broad linear symmetries.

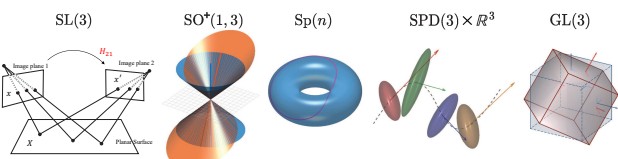

*Figure 1.* Examples of Lie groups and related manifolds in scientific applications. From left: the special linear group $\mathrm{SL}(3)$ (image homography), the Lorentz group $\mathrm{SO}^+(1,3)$ (spacetime symmetry), symplectic groups $\mathrm{Sp}(n)$ (Hamiltonian mechanics), the $\mathrm{SPD}(3) \times \mathbb{R}^3$ state space (probabilistic estimation), and the general linear group $\mathrm{GL}(3)$ (modeling stress-strain in continuum mechanics).

## 1. Introduction

Exploiting symmetries in data is a fundamental principle of geometric deep learning. By enforcing equivariance, where model outputs transform predictably with inputs, neural networks can leverage strong inductive biases to improve

*Equal contribution [1]University of Michigan, Ann Arbor, MI, USA [2]University of Pennsylvania, Philadelphia, PA, USA [3]Massachusetts Institute of Technology, Cambridge, MA, USA. Correspondence to: Chankyo Kim <chankyo@umich.edu>.

*Proceedings of the 43$^{rd}$ International Conference on Machine Learning*, Seoul, South Korea. PMLR 306, 2026. Copyright 2026 by the author(s).

data efficiency and generalization.

Substantial progress has been made for compact symmetry groups such as rotations $\mathrm{SO}(3)$ and Euclidean isometries $\mathrm{E}(n)$ (Bronstein et al., 2021; Cohen & Welling, 2016; Thomas et al., 2018; Satorras et al., 2021; Geiger & Smidt, 2022). However, many geometric and physical systems exhibit broader *linear* symmetries that are naturally expressed by the non-compact general linear group $\mathrm{GL}(n) = \{A \in \mathbb{R}^{n \times n} : \det(A) \neq 0\}$. Despite its ubiquity, general-purpose, exact equivariant architectures for $\mathrm{GL}(n)$ remain comparatively underdeveloped. Appendix D summarizes these groups and applications.

A key challenge for $\mathrm{GL}(n)$-equivariant learning is that the obstruction is algebraic, not topological. Semisimple groups, whether compact (e.g., $\mathrm{SO}(3)$) or non-compact (e.g., $\mathrm{SL}(n, \mathbb{R})$, $\mathrm{SO}^+(1,3)$), admit a non-degenerate Ad-invariant Killing form, and this form underlies the invariant contractions, gating, and normalization used by prior steerable and Lie-algebraic architectures (Thomas et al., 2018; Geiger & Smidt, 2022; Lin et al., 2024a). The Lie algebra $\mathfrak{gl}(n) = \mathbb{R}I \oplus \mathfrak{sl}(n)$ is reductive but *not* semisimple: the center $\mathbb{R}I$ satisfies $\mathrm{ad}_{\lambda I} \equiv 0$, so the Killing form vanishes identically on it. Any invariant constructed from the Killing form is therefore *blind to the center*—two features differing only by a scalar multiple of the identity yield the same invariant scalar, and the corresponding nonlinear gates collapse to identity along that direction. As a result, existing approaches either restrict attention to semisimple algebras or rely on ad hoc choices that do not provide a stable, general recipe for $\mathrm{GL}(n)$ adjoint equivariance. Appendix D summarizes these groups and applications.

Beyond this algebraic obstacle, real-world geometric inputs are often *heterogeneous*, with different attributes obeying different transformation laws. Vectors transform by a left action (e.g., $v \mapsto Av$), whereas matrix-valued tensors such as uncertainty, inertia, or shape covariances transform by congruence (e.g., $\Sigma \mapsto A\Sigma A^\top$); under orthogonal frame changes the two laws coincide with adjoint conjugation, so a single Ad-equivariant backbone can process both. More broadly, similarity is the native transformation law for objects such as linear dynamics under basis change, velocity gradients, and stress tensors, where adjoint equivariance is the appropriate notion already at the full $\mathrm{GL}(n)$ level. Learning to *jointly* couple vector–matrix channels in a symmetry-consistent and numerically stable way remains challenging: naive flattening discards tensor structure, and eigen/spectral parameterizations can be ambiguous and fragile under differentiation (Magnus, 1985). These issues arise in state estimation, where velocities and covariance must be processed together, and in 3D Gaussian Splatting, which couples a mean $\mu \in \mathbb{R}^3$ with an anisotropic covariance $\Sigma \in \mathrm{SPD}(3)$.

In this paper, we introduce *Reductive Lie Neurons (ReLNs)*, a general-purpose architecture that is *exactly* equivariant to the adjoint action of $\mathrm{GL}(n)$ on $\mathfrak{gl}(n)$. ReLNs are built around a non-degenerate Ad-invariant bilinear form $\widetilde{B}$ on $\mathfrak{gl}(n)$ that resolves the Killing-form degeneracy on reductive algebras and supplies the invariant scalars used by every gate, pooling score, and readout in the network. As $\widetilde{B}$ remains non-degenerate on the center, ReLNs are *center-sensitive*: they distinguish trace and scale signals that semisimple architectures discard, while reducing exactly to the Killing form on the traceless ideal and recovering Lie Neurons (Lin et al., 2024a) as a special case. We evaluate ReLNs in four complementary regimes: a native $\mathrm{GL}(n)$ system-identification task, center-sensitive matrix-input tasks (uncertainty-aware drone dynamics, 3D Gaussian Splat representation learning), semisimple algebraic benchmarks ($\mathfrak{sl}(3)$, $\mathfrak{sp}(4)$, additional Lorentz top-tagging), and the EMLP double-pendulum benchmark.

Our main contributions are:

1. We propose **Reductive Lie Neurons (ReLNs)**, a novel, general-purpose architecture for exact $\mathrm{GL}(n)$ adjoint equivariance, built upon a Ad-invariant bilinear form that resolves the degeneracy issues of the standard Killing form on reductive algebras.

2. We introduce a **unified lifting** that embeds heterogeneous geometric inputs, left-acting vectors and congruence-transforming matrices (covariances, 3D Gaussian parameters), into a shared $\mathfrak{gl}(n)$ feature space, so a **single adjoint-equivariant backbone** processes them without per-symmetry redesign.

3. We demonstrate **practical utility across distinct**

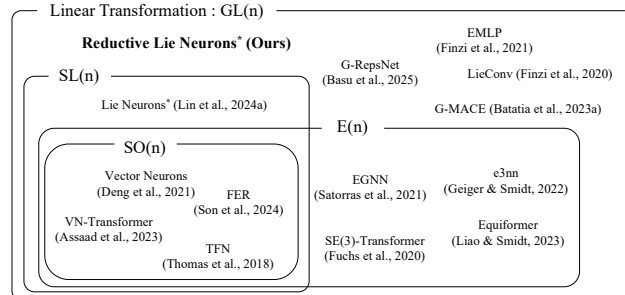

*Figure 2.* A taxonomy of selected representative equivariant neural architectures, categorized by the symmetries to which they are equivariant. This diagram situates our work, ReLNs, among other notable methods that are often specialized for subgroups such as $\mathrm{SL}(n)$, $\mathrm{SO}(n)$, or the Euclidean group $\mathrm{E}(n)$. An asterisk (*) denotes methods equivariant to the group's adjoint action. We include $\mathrm{E}(n)$ for completeness, noting that it is an affine extension whose translation lies outside the linear setting addressed here.

**regimes**: a native $\mathrm{GL}(n)$ system-identification task that requires center-sensitivity; center-sensitive matrix-input tasks (uncertainty-aware drone dynamics and 3D Gaussian Splat representation learning) where ReLNs improve over semisimple baselines; semisimple benchmarks ($\mathfrak{sl}(3)$, $\mathfrak{sp}(4)$, Lorentz top-tagging) where ReLNs match specialized prior work; and the EMLP double-pendulum benchmark where ReLNs match accuracy while substantially reducing per-step compute.

## 2. Related Work

Encoding symmetry into neural architectures is a well-established inductive bias for improving data efficiency and generalization (Bronstein et al., 2021). Most geometric deep learning work targets Euclidean isometries such as rotations and rigid motions (e.g., $\mathrm{SO}(n)$ and $\mathrm{SE}(n)$). For grid data, representative approaches include group-equivariant and steerable CNNs (Cohen & Welling, 2016; Weiler et al., 2018; Weiler & Cesa, 2021), with homogeneous-space formulations unifying many group-equivariant CNN constructions (Cohen et al., 2019). For sets and graphs, many methods build features from irreducible representations and tensorial message passing, including TFNs, $\mathrm{E}(n)$-GNNs, and equivariant transformers (Thomas et al., 2018; Fuchs et al., 2020; Satorras et al., 2021; Batatia et al., 2022; Liao & Smidt, 2023; Assaad et al., 2023; Hutchinson et al., 2021), alongside lighter vector-based variants (Deng et al., 2021; Son et al., 2024). Related theoretical work studies universality of invariant/equivariant architectures (Maron et al., 2019), and model-agnostic strategies such as canonicalization and frame averaging provide complementary routes to equivariance (Puny et al., 2022; Lin et al., 2024b; Kaba et al., 2023; Panigrahi & Mondal, 2024). While sev-

eral frameworks can represent higher-order tensors (e.g., TFN (Thomas et al., 2018) and EMLP (Finzi et al., 2021)), scalable architectural design for *general matrix-valued* quantities (e.g., covariances transforming as $\Sigma \mapsto R\Sigma R^\top$) remains less standardized in practice.

**Non-compact groups and** $\mathrm{GL}(n)$**.** Extending equivariance to non-compact symmetries remains an active area, notably exemplified by Lorentz-equivariant architectures that operate on Minkowski geometry in particle physics (Bogatskiy et al., 2020; Gong et al., 2022; Batatia et al., 2023; Zhdanov et al., 2024). Complementary research utilizes geometric (Clifford) algebras to represent features as multivectors, leading to frameworks such as Geometric Algebra Transformers (GATr) (Brehmer et al., 2023), Clifford-equivariant simplicial message passing (Liu et al., 2024), and lightweight designs like GLGENN (Filimoshina & Shirokov, 2025). Approaches to non-compact groups include generalizations of group convolutions and kernels (Xu et al., 2022; Helwig et al., 2023), constructions based on matrix functions or reductive groups (Batatia et al., 2024; 2023), Lie-algebra-based kernels and decompositions (Finzi et al., 2020; Mironenco & Forré, 2024; Shumaylov et al., 2025), differential-operator and differential-invariant methods (He et al., 2022; Shen et al., 2020; Jenner & Weiler, 2022; Sangalli et al., 2022; Li et al., 2024), and algebraic constraint-solving frameworks such as EMLP and G-RepsNet (Finzi et al., 2021; Basu et al., 2025). These lines provide general tools but can incur substantial overhead (e.g., integration/basis computation) or lack inductive biases such as locality. Within Lie-algebraic adjoint-equivariant learning, Lie Neurons (Lin et al., 2024a) establish an $\mathrm{Ad}$-equivariant neural network using the Killing form, but rely on the Killing form being non-degenerate and therefore do not directly cover reductive algebras such as $\mathfrak{gl}(n)$, on which the Killing form vanishes along the center. This gap is relevant when inputs include general matrix-valued quantities whose central component carries physically meaningful information (e.g., trace of a covariance, divergence of a flow). ReLNs supply a non-degenerate $\mathrm{Ad}$-invariant completion of the Killing form that agrees with it on the semisimple ideal while pairing the center, recovering Lie Neurons as a special case. Classical invariant filtering explicitly respects these transformation rules (Barrau & Bonnabel, 2016; Hartley et al., 2020), but is model-based and does not provide a learned backbone.

**Learning on 3D Gaussian splats.** 3D Gaussian Splatting represents geometry using anisotropic primitives parameterized by a mean $\mu \in \mathbb{R}^3$ and covariance $\Sigma \in \mathrm{SPD}(3)$. Recent learning frameworks over such primitives, including self-supervised learning and dynamics/segmentation models (Ma et al., 2025; Lu et al., 2024; Zhang et al., 2024; Ye et al., 2024), often treat heterogeneous attributes as loosely

coupled channels (e.g., concatenating $\mu$ with rotation parameters and applying generic architectures), which does not explicitly enforce the coupled transformation of $(\mu, \Sigma)$. This decoupled treatment disregards the underlying geometry, representing vectors and tensors as mixed latent feature that fail to recognize their transformation laws. Consequently, such models struggle to capture the symmetry of the scene without massive data augmentation. ReLN formulation targets this gap by embedding these heterogeneous attributes into a unified Lie-algebraic space, enforcing the coupled equivariant structure of $(\mu, \Sigma)$, exploiting the isomorphism between congruence and adjoint actions under SO(3).

ReLNs overcome these limitations by establishing exact adjoint equivariance on $\mathfrak{gl}(n)$ via a non-degenerate $\mathrm{Ad}$-invariant bilinear form. This approach bypasses the reliance on degenerate invariants and group-specific integration required by previous methods. Figure 2 situates ReLNs relative to prior equivariant frameworks.

## 3. Preliminaries

We build equivariant networks on a reductive Lie algebra $\mathfrak{gl}(n)$, requiring equivariance under the adjoint action $\mathrm{Ad}_g(X) = gXg^{-1}$ for $g \in \mathrm{GL}(n)$. A key obstruction is that $\mathfrak{gl}(n)$ is reductive but not semisimple: its canonical $\mathrm{Ad}$-invariant *symmetric bilinear form* (the Killing form) is degenerate. Many equivariant architectures rely on a *non-degenerate* invariant form to produce invariant scalars (e.g., gating/normalization) and to parameterize genuinely nonlinear equivariant operations (e.g., Vector Neurons (Deng et al., 2021), Lie Neurons (Lin et al., 2024a)); degeneracy makes these constructions ill-conditioned or effectively linear. We resolve this by introducing a non-degenerate $\mathrm{Ad}$-invariant bilinear form on $\mathfrak{gl}(n)$, enabling fully nonlinear adjoint-equivariant layers. Our scope is reductive Lie algebras; extending to non-reductive cases (e.g., $\mathfrak{aff}(n)$) generally requires additional non-canonical completions. For details of Lie theory and background, see Appendix A.

## 4. Reductive Lie Neurons: Architecture

ReLNs are built in three pieces. Section 4.1 introduces a non-degenerate $\mathrm{Ad}$-invariant bilinear form $\widetilde{B}$ that replaces the degenerate Killing form on reductive algebras. Section 4.2 defines the equivariant layer toolbox, with all invariant scalars expressed as $\widetilde{B}$-contractions. Section 4.3 describes the lifting of heterogeneous inputs into a shared $\mathfrak{gl}(n)$ feature space.

### 4.1. An $\mathrm{Ad}$-invariant bilinear form for reductive Lie algebras

A key obstruction to extending semisimple Lie-algebraic designs (Lin et al., 2024a) from $\mathfrak{sl}(n)$ to $\mathfrak{gl}(n)$ is that the

Killing form may be *degenerate* on non-semisimple algebras. A finite-dimensional Lie algebra $\mathfrak{g}$ is *reductive* if it decomposes as $\mathfrak{g} = \mathfrak{z}(\mathfrak{g}) \oplus [\mathfrak{g}, \mathfrak{g}]$ with $[\mathfrak{g}, \mathfrak{g}]$ semisimple, where the center $\mathfrak{z}(\mathfrak{g}) = \{Z \in \mathfrak{g} : [Z, X] = 0 \text{ for all } X \in \mathfrak{g}\}$ consists of elements with trivial adjoint action. The Killing form $B_{\mathfrak{g}}(X, Y) = \mathrm{Tr}(\mathrm{ad}_X \circ \mathrm{ad}_Y)$ vanishes identically on $\mathfrak{z}(\mathfrak{g})$: if $Z \in \mathfrak{z}(\mathfrak{g})$ then $\mathrm{ad}_Z = 0$, so $B_{\mathfrak{g}}(Z, \cdot) \equiv 0$. See Appendix A.4 for detailed definitions and the explicit form of $B_{\mathfrak{gl}(n)}$.

**Center-blindness of Killing-form invariants.** For $\mathfrak{gl}(n) = \mathbb{R}I \oplus \mathfrak{sl}(n)$ and any $X_1, Y \in \mathfrak{gl}(n)$, $\lambda \in \mathbb{R}$, setting $X_2 = X_1 + \lambda I$ and applying the closed form $B_{\mathfrak{gl}(n)}(X, Y) = 2n \, \mathrm{Tr}(XY) - 2 \, \mathrm{Tr}(X)\mathrm{Tr}(Y)$ (Eq. 20) gives $B_{\mathfrak{gl}(n)}(X_2, Y) = B_{\mathfrak{gl}(n)}(X_1, Y)$. Killing-form-based gates, pooling scores, and readouts thus treat $X_1$ and $X_2$ identically, and nonlinearities built from such invariants reduce to identity along the central direction. For matrix-valued inputs such as covariances and velocity gradients, the central component encodes physical information (uncertainty scale, divergence of a flow); a Killing-form-only architecture discards it by construction. We resolve this by constructing a non-degenerate $\mathrm{Ad}$-invariant completion of the Killing form on any reductive Lie algebra.

**Definition 4.1** (Modified Bilinear Form on a Reductive Lie Algebra). *If $\mathfrak{g}$ is reductive, then $\mathfrak{g} = \mathfrak{z}(\mathfrak{g}) \oplus [\mathfrak{g}, \mathfrak{g}]$, where $\mathfrak{z}(\mathfrak{g})$ is the center. Choose any $\mathrm{Ad}$-invariant inner product $\langle \cdot, \cdot \rangle_{\mathfrak{z}}$ on $\mathfrak{z}(\mathfrak{g})$ (for connected $G$ this is automatic since $\mathrm{Ad}|_{\mathfrak{z}(\mathfrak{g})} : G \to \mathrm{GL}(\mathfrak{z}(\mathfrak{g}))$ is locally constant; see Appendix A.2 for the formal definition of $\mathrm{Ad}$). For $Z_i \in \mathfrak{z}(\mathfrak{g})$ and $X_i \in [\mathfrak{g}, \mathfrak{g}]$ define*

$$\widetilde{B}(Z_1 + X_1, \, Z_2 + X_2) := \langle Z_1, Z_2 \rangle_{\mathfrak{z}} + B(X_1, X_2), \quad (1)$$

*where $B$ denotes the Killing form on the semisimple ideal $[\mathfrak{g}, \mathfrak{g}]$.*

**Proposition 4.2** (Non-degeneracy and $\mathrm{Ad}$-invariance). *The form $\widetilde{B}$ is symmetric, $\mathrm{Ad}$-invariant, and non-degenerate. Moreover, $\mathfrak{z}(\mathfrak{g})$ and $[\mathfrak{g}, \mathfrak{g}]$ are $\widetilde{B}$-orthogonal, with $\widetilde{B}|_{[\mathfrak{g},\mathfrak{g}]} = B$ and $\widetilde{B}|_{\mathfrak{z}(\mathfrak{g})} = \langle \cdot, \cdot \rangle_{\mathfrak{z}}$.*

*Proof sketch.* $B$ vanishes on $\mathfrak{z}(\mathfrak{g})$ and is $\mathrm{Ad}$-invariant; by construction $\langle \cdot, \cdot \rangle_{\mathfrak{z}}$ is $\mathrm{Ad}$-invariant. Symmetry is immediate. Nondegeneracy follows since $B$ is nondegenerate on the semisimple ideal and $\langle \cdot, \cdot \rangle_{\mathfrak{z}}$ is nondegenerate on the center; orthogonality holds because $B(\mathfrak{z}(\mathfrak{g}), [\mathfrak{g}, \mathfrak{g}]) = 0$. Detailed proofs are in Appendix C. $\square$

For our primary case $\mathfrak{g} = \mathfrak{gl}(n) = \mathbb{R}I \oplus \mathfrak{sl}(n)$, we use the closed form

$$\widetilde{B}(X, Y) = 2n \, \mathrm{Tr}(XY) - \mathrm{Tr}(X)\mathrm{Tr}(Y). \quad (2)$$

This $\widetilde{B}$ is our basic $\mathrm{Ad}$-invariant contraction throughout the ReLN architecture.

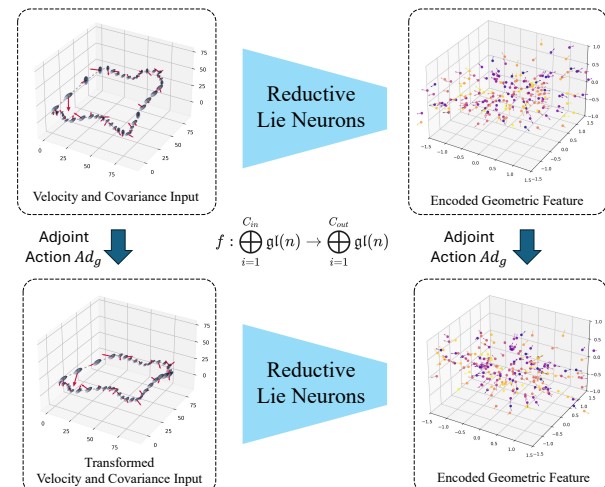

*Figure 3.* Adjoint equivariance using a unified representation for diverse geometric inputs. Our framework embeds inputs with different transformation rules, such as velocity ($v \mapsto Rv$) and covariance ($\Sigma \mapsto R\Sigma R^T$), into a common Lie algebra. Therefore, they transform under the same adjoint action $\mathrm{Ad}_g$, with which our network $f$ commutes as shown in the diagram.

**Verification and relation to prior bilinear forms.** Our concrete form in Eq. 2 satisfies the conditions of Definition 4.1. Decomposing a matrix $X = X_0 + \frac{1}{n}\mathrm{tr}(X)I$ (with $X_0 \in \mathfrak{sl}(n)$) reveals that our form separates orthogonally:

$$\widetilde{B}(X, Y) = \underbrace{2n \cdot \mathrm{tr}(X_0 Y_0)}_{B_{\mathfrak{sl}(n)}(X_0, Y_0)} + \underbrace{\mathrm{tr}(X)\mathrm{tr}(Y)}_{\text{inner product on } \mathbb{R}I} . \quad (3)$$

The first term is the Killing form on the semisimple ideal, used in Lie Neurons (Lin et al., 2024a). The second is an inner product on the center, which, under the isomorphism $\mathfrak{so}(3) \simeq \mathbb{R}^3$, recovers (up to a fixed global scaling) the dot product used in Vector Neurons (Deng et al., 2021). For $\mathfrak{so}(3)$, the hat map intertwines the vector action and the adjoint action (Lemma B.7), so $\mathrm{Ad}$-equivariance on $\mathfrak{so}(3)$ corresponds to standard $\mathrm{SO}(3)$-equivariance on $\mathbb{R}^3$. Our single form thus unifies these approaches and extends to the full reductive algebra $\mathfrak{gl}(n)$ (Appendix B).

**On the choice of inner product on the center.** Definition 4.1 allows any $\mathrm{Ad}$-invariant inner product $\langle \cdot, \cdot \rangle_{\mathfrak{z}}$ on $\mathfrak{z}(\mathfrak{g})$. For $\mathfrak{gl}(n)$ the center is one-dimensional, so all such choices differ by a positive scalar and are absorbed by adjacent learnable maps; we confirm this empirically in Appendix H.4, where varying the center scale by two orders of magnitude leaves performance unchanged. For reductive algebras with higher-dimensional centers, non-equivalent $\mathrm{Ad}$-invariant choices may induce different inductive biases; we leave this to future work. Our specific form (2) is the simplest non-degenerate $\mathrm{Ad}$-invariant extension of the Killing form that agrees with $B$ on $\mathfrak{sl}(n)$.

## 4.2. The ReLN Layer Toolbox

We represent multi-channel input as $x \in \mathbb{R}^{K \times C}$, where each column $x_c \in \mathbb{R}^K$ corresponds to a matrix $X_c \in \mathfrak{g}$ (via the vee/hat isomorphism, Appendix A).

**ReLN-Linear.** A linear map applied to the channel dimension $f(x; W) = xW$ with $W \in \mathbb{R}^{C \times C'}$ is equivariant: the group acts on the left (geometric dimension) while $W$ acts on the right (channel dimension), and thus these operations commute (formal proof in Appendix E).

**Equivariant Nonlinearities.** We introduce two complementary nonlinear primitives; full definitions, parameterizations, and stability prescriptions are in Appendix E.

- **ReLN-ReLU**: To avoid basis-dependent elementwise activations, we use $\widetilde{B}$ to build an *invariant scalar gate* and apply it along an *equivariant direction*. For input features $x \in \mathbb{R}^{K \times C}$, we define an equivariant reference direction $d = xW$, where $W \in \mathbb{R}^{C \times C}$ is a learnable weight matrix. The feature is updated by

$$x'_c = \begin{cases} x_c, & \text{if } \widetilde{B}(x_c^\wedge, d_c^\wedge) \leq 0 \\ x_c + \widetilde{B}(x_c^\wedge, d_c^\wedge)\, d_c, & \text{otherwise} \end{cases} \quad (4)$$

where $x_c^\wedge, d_c^\wedge \in \mathfrak{g}$ denote the matrix forms. Since $\widetilde{B}(\cdot, \cdot)$ is $\mathrm{Ad}$-invariant the gating condition is invariant, and since $d$ is an equivariant linear map of $x$ the vector update is strictly equivariant.

- **ReLN-Bracket.** Following prior work (Lin et al., 2024a), we include a layer that leverages the Lie bracket (matrix commutator). This operation is an $\mathrm{Ad}$-equivariant primitive that creates nonlinear interactions by measuring the non-commutativity. The layer applies two independent linear maps, parameterized by weights $W_a, W_b \in \mathbb{R}^{C \times C}$, to the input channels $x_{\mathrm{in}}$ to produce two features, computes their commutator, and injects the vectorized result as a shared residual:

$$x_{\mathrm{out}} = x_{\mathrm{in}} + \left( \left[ (x_{\mathrm{in}} W_a)^\wedge, (x_{\mathrm{in}} W_b)^\wedge \right] \right)^\vee. \quad (5)$$

**Equivariant Pooling and Invariant Layers.**

- **Max-Killing Pooling.** Given an unordered set $\{x_n\}_{n=1}^N$ with $x_n \in \mathbb{R}^{K \times C}$, we form a *dynamic, per-element query* $d_n = x_n W_q$ via a shared ReLN-Linear map with $W_q \in \mathbb{R}^{C \times C}$. Because $W_q$ acts only on channels, $d_n$ transforms equivariantly with $x_n$. For each channel $c$, we select $n^*(c) = \arg\max_n \widetilde{B}(X_{n,c}, (d_n)_c^\wedge)$ and return $X_c^{\max} = X_{n^*(c),c}$. The $\widetilde{B}$-score is $\mathrm{Ad}$-invariant, parameter sharing across $n$ together with the symmetric $\arg\max$ ensures permutation invariance, and the gathered feature is $\mathrm{Ad}$-equivariant.

- **Invariant Readout.** To produce a group-invariant output, this layer contracts a feature $X_c$ with itself: $y_c = \widetilde{B}(X_c, X_c)$, which is $\mathrm{Ad}$-invariant by construction.

## 4.3. Unified Lifting of Heterogeneous Geometric Inputs

Because ReLN layers operate on matrix representatives in $\mathfrak{gl}(n)$, inputs with different transformation laws can be embedded into a shared feature space and processed by the same adjoint-equivariant backbone. We use three liftings throughout our experiments:

- **Left-acting 3D vectors** $v \in \mathbb{R}^3$ (velocities, positions): the hat map $v \mapsto v^\wedge \in \mathfrak{so}(3) \subset \mathfrak{gl}(3)$ intertwines the rotation action with adjoint conjugation, $\widehat{Rv} = R\hat{v}R^\top = \mathrm{Ad}_R(\hat{v})$ for $R \in \mathrm{SO}(3)$ (Lemma B.7). For 4-momenta under the Lorentz group $\mathrm{SO}^+(1,3)$, an analogous block embedding into $\mathfrak{gl}(5)$ achieves the same intertwining property (Appendix I.2).

- **Congruence-transforming matrices** $C \in \mathrm{SPD}(n)$ (covariances): under orthogonal frame changes, congruence $C \mapsto RCR^\top$ coincides with adjoint conjugation. We treat $C$, or $\log C \in \mathrm{Sym}(n) \subset \mathfrak{gl}(n)$ when scale varies multiplicatively, as a direct $\mathfrak{gl}(n)$ feature.

- **Similarity-transforming operators** $A \in \mathfrak{gl}(n)$ (linear dynamics, velocity gradients, stress tensors): the native transformation law $A \mapsto SAS^{-1}$ for $S \in \mathrm{GL}(n)$ is the adjoint conjugation on $\mathfrak{gl}(n)$, so $A$ is supplied as a direct feature.

## 5. Experiments

Each benchmark targets a distinct aspect of ReLNs. (a) *Native full-*$\mathrm{GL}(n)$ tasks (System Identification, Section 5.1) test reductive equivariance where both inputs and task symmetry require the center, where semisimple Lie-algebraic baselines are insufficient by construction. (b) *Center-sensitive matrix-input tasks* (drone dynamics, Section 5.3; 3D Gaussian splats, Section 5.4) test reductive inputs under orthogonal task symmetry, where semisimple architectures discard the central component while ReLNs preserve it. (c) *Semisimple benchmarks* ($\mathfrak{sl}(3)$ Platonic, Section 5.2.1; $\mathfrak{sp}(4)$ invariant regression, Section 5.2.2; Lorentz top-tagging, Appendix I) verify backward compatibility: in the absence of a nontrivial center, ReLNs should match specialized prior work. (d) The *EMLP double-pendulum benchmark* (Section 5.5) evaluates efficiency relative to numerical constraint-solving benchmark across multiple symmetry groups. Across all settings we compare against non-equivariant baselines (MLP, ResNet), Lie-algebraic equivariant models (Lie Neurons), and strong equivariant architectures including Vector Neurons (VN), Tensor Field Networks (TFN), and an $\mathrm{SE}(3)$-Transformer.

*Table 1.* System identification under similarity transforms. *Trace MSE* measures center recovery; *Canonical MSE* measures intrinsic similarity-invariant error. Means $\pm$ std over 3 seeds.

| Method | Split | Trace MSE $\downarrow$ | Canonical MSE $\downarrow$ |
|---|---|---|---|
| Avg-LS | All | $0.0211 \pm 0.0004$ | $0.0044 \pm 0.0002$ |
| MLP | ID | $0.0145 \pm 0.0005$ | $0.0041 \pm 0.0002$ |
| | GL | $0.0489 \pm 0.0064$ | $0.1480 \pm 0.0220$ |
| | GL-Hard | $1.0285 \pm 0.0270$ | $57.4099 \pm 7.3559$ |
| Lie Neurons (Lin et al., 2024a) | All | $1.6850 \pm 0.0500$ | $0.0658 \pm 0.0017$ |
| **ReLN (Ours)** | All | $\mathbf{0.0133 \pm 0.0005}$ | $\mathbf{0.0039 \pm 0.0002}$ |

*Table 2.* Platonic solid classification (mean $\pm$ std over 5 runs). ID = in-distribution; RC = rotated-camera (500 random $\mathrm{SO}(3)$ test rotations). "+Aug" denotes training with random $\mathrm{SO}(3)$ augmentation of input homographies. Higher is better ($\uparrow$).

| Model | # Params | ID Acc (mean $\pm$ std) | RC Acc (mean $\pm$ std) |
|---|---|---|---|
| MLP | 206,339 | $95.76\% \pm 0.65\%$ | $36.54\% \pm 0.99\%$ |
| MLP + Aug | 206,339 | $81.47\% \pm 0.77\%$ | $81.20\% \pm 2.34\%$ |
| MLP (wider) | 411,479 | $96.82\% \pm 0.53\%$ | $36.55\% \pm 0.34\%$ |
| MLP (wider) + Aug | 411,479 | $85.22\% \pm 1.46\%$ | $83.43\% \pm 0.51\%$ |
| Lie Neurons | 331,272 | $99.62\% \pm 0.25\%$ | $99.61\% \pm 0.14\%$ |
| **ReLN (Ours)** | 331,272 | $\mathbf{99.78\% \pm 0.04\%}$ | $\mathbf{99.78\% \pm 0.04\%}$ |

## 5.1. Native $\mathrm{GL}(n)$ System Identification

To test full $\mathrm{GL}(n)$ adjoint equivariance, we recover an unknown linear dynamics matrix $A \in \mathfrak{gl}(3)$ of a system $z_{t+1} = Az_t$ from a set of noisy windowed least-squares estimates $\{A_{ij}\}$. Under a basis change $z \mapsto Sz$ with $S \in \mathrm{GL}(3)$, both inputs and target transform by similarity $(A \mapsto SAS^{-1})$, which is the adjoint action on $\mathfrak{gl}(3)$. Crucially, $\mathrm{Tr}(A)$ governs global expansion/contraction and lives in the center, so recovering $A$ requires *both* the semisimple ideal and the center. We compare against an MLP, Lie Neurons, and a structured *Avg-LS* baseline that averages the windowed estimates, evaluating on three test conditions: **ID** (training basis), **GL** (random $S$ at test time), and **GL-Hard** (ill-conditioned $S$). We report *Trace MSE* (center recovery) and *Canonical MSE* (intrinsic similarity-invariant error). Full details are in Appendix H.

**Results.** Table 1 shows three regimes. The MLP collapses under test-time similarity (57.4 Canonical MSE on GL-Hard), confirming that training does not absorb arbitrary basis changes. Lie Neurons is stable in Canonical MSE but its Trace MSE is two orders of magnitude worse than ReLN (1.685 vs. 0.013): Killing-form-only invariants are blind to $\mathrm{Tr}(A)$ by construction. ReLN attains the lowest error on both metrics simultaneously, also outperforming the structured Avg-LS baseline. The reductive completion is thus both *necessary*, semisimple architectures cannot recover the trace, and *sufficient*, ReLN does so while preserving similarity equivariance.

## 5.2. Algebraic Benchmarks on Semisimple Lie Algebras

To verify that our general $\mathfrak{gl}(n)$ framework correctly generalizes to semisimple subalgebras, we evaluate ReLN on two Lie-algebraic benchmarks introduced by Lie Neurons (Lin et al., 2024a).

### 5.2.1. PLATONIC SOLID CLASSIFICATION

We validate our model on the Platonic solid classification benchmark from (Lin et al., 2024a), testing adjoint equivariance where camera rotations induce a conjugation action on the homographies between projected faces of the solid, represented as $\mathrm{SL}(3)$. Full experimental details are in Appendix G.2.

The results in Table 2 show that non-equivariant baselines fail to generalize to rotated camera views; this gap persists even with augmentation or doubled capacity. ReLN achieves near-perfect accuracy across both splits and matches Lie Neurons within run-to-run variability. This confirms that our general $\mathfrak{gl}(n)$ framework operates effectively on semisimple subalgebras: built-in adjoint equivariance on the parent group yields robust behavior when restricted to subgroups like $\mathrm{SO}(3)$ and $\mathrm{SL}(3)$.

### 5.2.2. INVARIANT FUNCTION REGRESSION ON $\mathfrak{sp}(4)$.

To probe algebraic generality beyond $\mathfrak{sl}(n)$, we regress a highly nonlinear $\mathrm{Sp}(4)$-invariant scalar on the real symplectic Lie algebra $\mathfrak{sp}(4, \mathbb{R})$ (the Lie algebra underlying Hamiltonian/symplectic symmetry). For $X, Y \in \mathfrak{sp}(4, \mathbb{R})$, the target is

$$g(X, Y) = \sin\big(\mathrm{Tr}(XY)\big) + \cos\big(\mathrm{Tr}(YY)\big)$$
$$- \tfrac{1}{2}\mathrm{Tr}(YY)^3 + \det(XY) + \exp\big(\mathrm{Tr}(XX)\big). \tag{6}$$

We sample 10k training and 10k test pairs and compare ReLN to MLP baselines (with/without $\mathrm{Sp}(4)$ conjugation augmentation) and Lie Neurons. We report test MSE, MSE averaged over 500 random adjoint actions, and the invariance error.

As shown in Table 3, non-equivariant MLPs are orders of magnitude less accurate and exhibit substantially larger invariance error, indicating that they fail to reliably capture the underlying group structure even with $\mathrm{Sp}(4)$ conjugation augmentation. In contrast, Lie Neurons and ReLN achieve low test MSE and near-zero invariance error, demonstrating stable invariant regression on $\mathfrak{sp}(4)$. At this parameter scale, the performance of the two equivariant models is comparable. These results support that our model matches the accuracy of specialized Lie-algebraic baselines while retaining a unified construction across diverse Lie algebras.

*Table 3.* Regression performance and invariance error on $\mathfrak{sp}(4)$. "Tr. Aug." indicates training augmentation.

| Model | Tr. Aug. | # Params | Test Aug. ID | Test Aug. SP(4) | Inv. Error |
|---|---|---|---|---|---|
| MLP 256 | Id | 137,217 | 0.126 | 1.360 | 0.722 |
| | SP(4) | 137,217 | 0.192 | 0.587 | 0.476 |
| MLP 512 | Id | 536,577 | 0.107 | 0.906 | 0.585 |
| | SP(4) | 536,577 | 0.123 | 0.446 | 0.374 |
| Lie Neurons | Id | 263,170 | $5.83 \times 10^{-4}$ | $5.84 \times 10^{-4}$ | $3.84 \times 10^{-7}$ |
| ReLN (ours) | Id | 263,170 | $5.14 \times 10^{-4}$ | $5.14 \times 10^{-4}$ | $4.73 \times 10^{-7}$ |

## 5.3. Drone State Estimation with Geometric Uncertainty

We evaluate ReLN on a drone state-estimation task that reconstructs 3D trajectories from noisy velocity measurements $\mathbf{v} \in \mathbb{R}^3$ and time-varying uncertainty covariances $C \in \mathrm{SPD}(3)$ during highly dynamic flights. This setup requires a model to jointly process vector and matrix data in a geometrically consistent and uncertainty-aware manner.

**Experimental Setup.** We use a large-scale synthetic dataset of 200 aggressive trajectories (over 13 hours of flight); details are in Appendix J. We evaluate ReLN against non-equivariant 1D ResNets, VN (Deng et al., 2021), Lie Neurons (Lin et al., 2024a), and two spherical-harmonics-based equivariant architectures that use steerable bases and tensor-product coupling: TFN (Thomas et al., 2018) and SE(3)-Transformers (Fuchs et al., 2020). To isolate the effect of uncertainty fusion, we test three variants for each equivariant model: velocity-only, velocity + covariance, and velocity + log-covariance. For baselines such as VN that cannot natively process matrix inputs, we implement an eigendecomposition-based strategy (Appendix J.3); for TFN and SE(3)-Transformer we apply irreducible representation (irrep) decomposition.

**Irrep decomposition vs. unified adjoint processing.** Steerable baselines (TFN, SE(3)-Transformer) represent $C$ through irreducible components (trace and traceless-symmetric parts, corresponding to $\ell = 0$ and $\ell = 2$), and learn interactions with $\mathbf{v}$ via tensor-product coupling between fibers. This decomposition separates the holistic geometry of the measurement–uncertainty pair, relying on tensor-product coupling to reconnect these components; see Appendix J.3 for the architecture and decomposition. In contrast, ReLN embeds both $\mathbf{v}$ and $C$ into a common matrix-valued representation with a single conjugation rule: $\mathbf{v}$ is lifted to $\mathfrak{so}(3) \subset \mathfrak{gl}(3)$ and $C$ (or $\log C$) is provided as a matrix feature. This yields a uniform $\mathrm{Ad}$-action for the pair and allows gating/normalization using the same $\mathrm{Ad}$-invariant form $\tilde{B}$. The final velocity estimate is obtained by projecting the network output onto its skew-symmetric component; implementation details are in Appendix J.

*Table 4.* Drone state estimation performance (ATE and RTE in meters). $v, C, \log C$ denote velocity, covariance, and log-covariance inputs. Standard deviations over 3 seeds for headline configurations are reported in Appendix J.6.

| Model | Input | ID ATE ↓ | ID RTE ↓ | SO(3) ATE ↓ | SO(3) RTE ↓ |
|---|---|---|---|---|---|
| *Non-Equiv.* | | | | | |
| ResNet | $(v, C)$ | 205.11 | 106.07 | 213.26 | 109.37 |
| *Equiv. Baselines* | | | | | |
| VN (Deng et al., 2021) | $v$ | 17.36 | 13.51 | 17.36 | 13.51 |
| VN (Deng et al., 2021) | $(v, C)$ | 191.78 | 98.39 | 190.22 | 98.26 |
| TFN (Thomas et al., 2018) | $(v, C)$ | 17.56 | 14.40 | 17.56 | 14.40 |
| SE(3)-Tr. (Fuchs et al., 2020) | $(v, C)$ | 21.67 | 16.77 | 21.67 | 16.77 |
| Lie Neurons (Lin et al., 2024a) | $(v, C)$ | 16.86 | 13.65 | 16.86 | 13.65 |
| Lie Neurons (Lin et al., 2024a) | $(v, \log)$ | 15.65 | 12.04 | 15.65 | 12.04 |
| *Our Equivariant Models* | | | | | |
| ReLN (Ours) | $v$ | 16.85 | 12.70 | 16.85 | 12.70 |
| ReLN (Ours) | $(v, C)$ | 16.49 | 13.02 | 16.49 | 13.02 |
| ReLN (Ours) | $(v, \log C)$ | **13.92** | **11.04** | **13.92** | **11.04** |
| ReLN (Ours) (no semisimple) | $(v, \log C)$ | 16.27 | 12.65 | 16.27 | 12.65 |

### 5.3.1. RESULTS AND ANALYSIS

Table 4 summarizes the main findings (full results in Appendix J.6).

**Equivariance is necessary for robustness and accuracy.** Non-equivariant ResNets exhibit large trajectory errors and do not benefit from adding covariance, indicating that concatenating matrix-valued uncertainty does not yield geometry-consistent fusion. Their errors remain high under test-time SO(3) rotations, confirming poor robustness to measurement-frame changes.

**The Interface for Covariance Matters.** Among equivariant baselines, velocity-only models already attain low error (e.g., VN), but performance becomes highly sensitive once covariance is introduced. The VN $(v, C)$ variant collapses toward non-equivariant performance (Table 4), consistent with the fact that an eigendecomposition-based interface separates coupled degrees of freedom (eigenvectors vs. eigenvalues) and yields an unstable signal for learning. By contrast, TFN and SE(3)-Transformer incorporate $(v, C)$ through steerable tensor representations, leading to moderate improvements.

**Local equivariant operators offer superior robustness for uncertainty fusion.** While SE(3)-Transformer shows a slight performance advantage in the velocity-only regime $(v)$, the introduction of covariance shifts the advantage to TFN, which consistently improves over SE(3)-Transformer across all covariance-integrated variants. This pattern suggests that attention mechanisms model global dependencies well in simple velocity sequences, while local convolutional aggregation provides a stronger and more stable inductive bias for processing high-frequency dynamics and uncertainty.

**ReLN yields the most consistent uncertainty-aware gains.** ReLN achieves the best overall performance, and

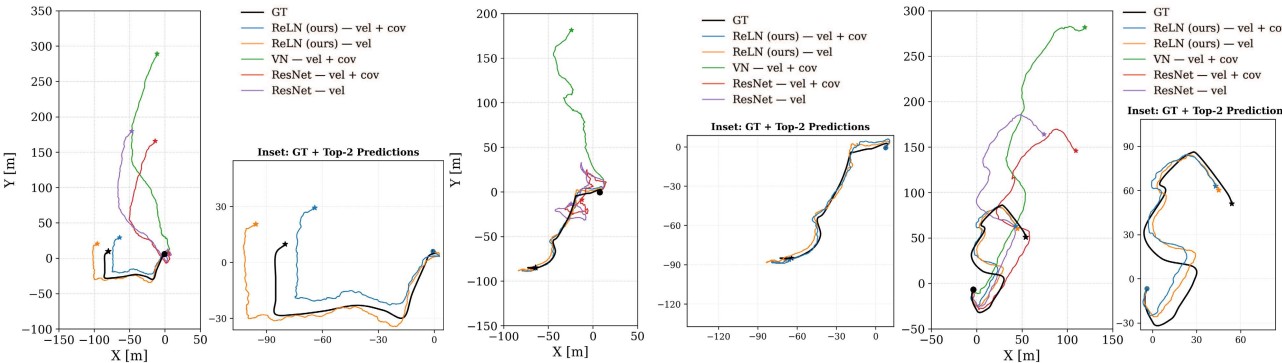

*Figure 4.* Trajectory reconstruction across flight difficulties: high (left), best-case (middle), and medium (right). ReLN models consistently track the ground truth (black) with high fidelity, especially when leveraging covariance. Insets provide a magnified view of the two best-performing variants (ReLN) to highlight their accuracy.

its strongest configuration uses $(v, \log C)$ (ATE 13.92, RTE 11.04). Unlike tensorial pipelines, ReLN represents velocity and covariance in a shared matrix-algebraic space with a single conjugation rule, enabling uncertainty-conditioned contractions and gating without violating equivariance. The improvement from $C$ to $\log C$ further supports that respecting the intrinsic geometry of SPD(3) via the matrix logarithm improves conditioning for learning.

**Ablative Validation of Reductive Decomposition.** ReLN improves over its *semisimple-only* variant (the no-center model, equivalent to Lie Neurons), indicating that incorporating the central component $\mathfrak{z}(\mathfrak{g})$ is beneficial for uncertainty-aware estimation. Conversely, removing the semisimple ideal (retaining only the center) degrades performance, increasing ATE from 13.92 to 16.27 (Table 4). Together, these ablations support that the best performance arises from modeling the *full* reductive structure.

**Discussion.** Overall, the results indicate that ReLNs act as a *geometry- and uncertainty-aware* estimator, remaining stable under random measurement-frame changes (test-time SO(3) rotations). While ReLN is not a classical recursive (Markovian) filter, it learns to fuse velocity sequences with their time-varying covariances through uncertainty-dependent modulation, improving trajectory reconstruction. This suggests ReLNs as a modular backbone for estimation pipelines that must handle matrix-valued uncertainty. A natural direction is to characterize when the learned fusion approximates uncertainty-adaptive integration, and to integrate ReLNs with classical state-estimation frameworks.

### 5.4. Equivariance for 3D Gaussian Splatting (3DGS)

Standard point cloud networks typically process only spatial coordinates. 3DGS, however, represents scenes using anisotropic 3D Gaussians parameterized by a mean position $\mu \in \mathbb{R}^3$ and a covariance matrix $\Sigma \in \mathrm{SPD}(3)$. This intro-

duces a geometric challenge: a global rotation $R \in \mathrm{SO}(3)$ acts differently on these components, where $\mu$ transforms as a vector ($R\mu$) and $\Sigma$ transforms by congruence ($R\Sigma R^\top$).

To evaluate ReLNs in this multi-modal geometric setting, we adapt the masked autoencoder framework of Ma et al. (2025), which establishes a self-supervised pretraining baseline for 3D Gaussian splats. While the original architecture processes Gaussian parameters as loosely coupled features using standard Transformers, we redesign the encoder and decoder using ReLN blocks to enforce equivariance through a shared $\mathfrak{gl}(3)$ adjoint representation, evaluated under rotations where covariance congruence coincides with adjoint conjugation. The overall architecture of our ReLN-integrated Gaussian-MAE is illustrated in Figure 7. Specifically, we unify these geometric types by embedding them into $\mathfrak{gl}(3)$: the mean position $\mu$ is treated as a translation, while the covariance $\Sigma$ is mapped to the linear subspace $\mathrm{Sym}(3) \subset \mathfrak{gl}(3)$ via the matrix logarithm. This ensures consistent geometric representations of 3D shapes under arbitrary orientations.

**Experimental Setup.** We utilize the ShapeNet dataset processed into 3D Gaussian primitives for self-supervised pretraining, following the protocol of Ma et al. (2025). The model is fine-tuned on ModelNet10 for classification. We compare against the baseline of Ma et al. (2025) and against a Lie Neurons backbone trained under the same protocol. Refer to Appendix L for full experimental details.

**Results.** Table 5 evaluates classification accuracy on aligned and randomly rotated ($0°$–$180°$) test sets. The non-equivariant baseline collapses from 93.39% to 18.28% under rotation. Lie Neurons achieves rotation robustness (92.20% on both splits) but underperforms on aligned data, consistent with semisimple invariants discarding the central component of $\Sigma$ that encodes uncertainty scale. ReLN preserves both equivariance and the center, attaining 94.82%

*Table 5.* **Rotation robustness on 3D Gaussian splats.** Classification accuracy (%) on ModelNet10 for the baseline (Ma et al., 2025) and ReLN. *Standard* uses the aligned test set; *Rotated* applies random rotations ($0°$–$180°$).

| Method | Standard Acc. (%) | Rotated Acc. (%) |
|---|---|---|
| Baseline (Ma et al., 2025) | 93.39 | 18.28 |
| Lie Neurons | 92.20 | 92.20 |
| **ReLN (Ours)** | **94.82** | **95.15** |

*Table 6.* Test rollout error on the EMLP double-pendulum benchmark. ReLN matches or improves upon EMLP across different symmetry groups ($O(2)$, $SO(2)$, $D_6$). Values represent the geometric mean of relative errors, with the standard deviation over 3 trials in parentheses; e.g., $0.012\,(2)$ denotes $0.012 \pm 0.002$.

| Metric | $O(2)$ | | $SO(2)$ | | $D_6$ | | MLP-HNNs |
|---|---|---|---|---|---|---|---|
| | EMLP | ReLN | EMLP | ReLN | EMLP | ReLN | |
| Rollout Error | 0.012 (2) | **0.011 (2)** | 0.015 (3) | **0.010 (4)** | 0.013 (2) | **0.011 (2)** | 0.028 |

aligned and $95.15\%$ rotated—a 0.33-point variation relative to the aligned setting. ReLN also converges in fewer epochs during pretraining and achieves lower reconstruction error across geometric attributes (rotation, scale; Appendix Figure 8). Together, these results support that enforcing the coupled transformation rules of $(\mu, \Sigma)$ via a reductive Lie-algebraic backbone improves both robustness to orientation changes and the quality of learned representations.

### 5.5. EMLP Double-Pendulum Benchmark and Efficiency

We evaluate ReLN on the EMLP double-pendulum benchmark (Finzi et al., 2021), which learns Hamiltonian dynamics under multiple symmetry groups ($O(2)$, $SO(2)$, and $D_6$). Following the standard protocol, we report test rollout errors in Table 6; additional details are in Appendix M.

**Accuracy.** ReLN matches or slightly improves EMLP across all symmetries (e.g., $SO(2)$: **0.010** vs. 0.015), without introducing group-specific architectural modifications.

**Compute and latency.** With matched hidden size and representation dimension, ReLN-HNN is substantially cheaper per step than EMLP-HNN (Table 7): 142,190 vs. 1,589,909 FLOPs ($11.18\times$) and 2.216 vs. 61.635 ms ($27.81\times$). This gap is consistent with ReLN using closed-form matrix operations from exact $\mathrm{Ad}$-equivariant primitives on $\mathfrak{gl}(n)$, whereas EMLP relies on symmetry-dependent basis construction and projection. Overall, these results support ReLN as a reusable equivariant backbone that attains EMLP-level accuracy while substantially reducing per-step computation across different symmetry choices.

### 6. Discussion and Limitations.

ReLNs are exactly $\mathrm{GL}(n)$-adjoint equivariant by construction, and this is the appropriate symmetry for inputs whose

*Table 7.* Computational cost on the HNN task. ReLN-HNN reduces FLOPs/step and inference latency relative to EMLP-HNN.

| Model | # Params | FLOPs / step | Inference (ms) |
|---|---|---|---|
| MLP-HNN | 34,817 | 70,400 | 0.159 |
| EMLP-HNN | 55,569 | 1,589,909 | 61.635 |
| **ReLN-HNN (Ours)** | 69,889 | **142,190** | **2.216** |

native transformation law is similarity—linear dynamics under basis change, velocity gradients, stress tensors. For covariance-like quantities, the physically standard transformation is congruence ($\Sigma \mapsto A\Sigma A^\top$), which coincides with adjoint conjugation ($\Sigma \mapsto A\Sigma A^{-1}$) only for orthogonal frame changes ($A^{-1} = A^\top$). Our unified Lie-algebraic interface is therefore most directly interpretable for covariance data under rigid frame changes; extending the same measurement–uncertainty coupling to general $\mathrm{GL}(n)$ coordinate changes requires additional geometric justification and may call for alternative liftings or invariants.

**Numerical considerations and theoretical guarantees.** On the theoretical side, while ReLNs benefit from the reduced hypothesis class implied by exact $\mathrm{GL}(n)$ equivariance, formal guarantees on sample complexity, convergence, and universal approximation within the class of continuous $\mathrm{GL}(n)$-adjoint-equivariant functions remain open.

### 7. Conclusion

We introduced Reductive Lie Neurons (ReLNs), an architecture exactly equivariant to the adjoint action of $\mathrm{GL}(n)$ on $\mathfrak{gl}(n)$. The construction resolves the Killing-form degeneracy on reductive Lie algebras via a non-degenerate $\mathrm{Ad}$-invariant bilinear form $\widetilde{B}$, and a single backbone processes heterogeneous geometric inputs through explicit Lie-algebraic liftings, removing the need for per-symmetry redesign. Empirically, ReLNs match specialized prior work on semisimple benchmarks, recover the global dynamics matrix in a native $\mathrm{GL}(n)$ system-identification task where semisimple methods fail by construction, and improve semisimple baselines on center-sensitive matrix-input tasks (drone state estimation, 3D Gaussian-splat representation learning).

Beyond reusable equivariant primitives, the framework offers a unified modeling perspective for problems whose features are matrix-valued and whose symmetry is the adjoint action: a single reductive feature space replaces a collection of group-specific designs. We expect these capabilities to benefit data-efficient representation learning in vision and robotics, particularly for structured 3D scene models such as 3D Gaussian splats and world models, where coupled transformation rules are central. Future work will scale ReLNs to larger physical systems and extend the framework to non-reductive symmetry groups.

## Impact Statement

This work presents a general equivariant learning framework to advance geometric machine learning. By enforcing geometric consistency, the proposed approach is applicable to downstream tasks such as robotics, 3D perception, and geometric representations including 3D Gaussian Splatting.

As a methodological contribution, this work does not raise direct ethical concerns. Any broader societal impact depends on downstream applications, which are subject to standard ethical and safety considerations.

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

# A. Lie-theoretic preliminaries

This appendix provides an overview of key concepts and derivations from Lie group theory relevant to our construction of $\mathrm{GL}(n)$ adjoint-equivariant neural networks.

## A.1. Lie groups, Lie algebras, hat/vee

A *matrix Lie group* $G \subset \mathrm{GL}(n)$ is a smooth subgroup of invertible matrices. Its Lie algebra $\mathfrak{g} = \mathrm{Lie}(G)$ is the tangent space at the identity and is identified with a subspace of $\mathfrak{gl}(n)$. Fix a basis $\{E_i\}_{i=1}^m$ of $\mathfrak{g}$. The coordinate maps are:

$$\wedge : \mathbb{R}^m \to \mathfrak{g}, \quad x = (x_i) \mapsto x^\wedge = \sum_i x_i E_i, \qquad \vee : \mathfrak{g} \to \mathbb{R}^m, \quad X \mapsto X^\vee. \tag{7}$$

These maps let us implement algebra-valued features as Euclidean vectors in code.

The associated Lie algebra $\mathfrak{g} = \mathrm{Lie}(G)$ is the tangent space at the identity element $e \in G$. It carries a bilinear, antisymmetric product called the *Lie bracket*, given by

$$[A, B] = AB - BA, \tag{8}$$

in the case of $\mathrm{GL}(n)$ which captures the infinitesimal structure of the group near the identity. The bracket quantifies non-commutativity of generators: $[A, B] = 0$ implies commutativity, whereas $[A, B] \neq 0$ indicates a non-trivial interaction.

## A.2. Representations and the adjoint action

A representation $\Phi : G \to \mathrm{GL}(V)$ differentiates to $\phi : \mathfrak{g} \to \mathfrak{gl}(V)$ by

$$\phi(X) = \left.\frac{d}{dt}\right|_{t=0} \Phi(\exp(tX)). \tag{9}$$

The adjoint representation $\mathrm{Ad} : G \to \mathrm{GL}(\mathfrak{g})$ is defined to be the differential of group conjugation at the identity

$$\mathrm{Ad}_g(X) = \left.\frac{d}{dt}\right|_{t=0} g(\exp(tX))g^{-1}. \tag{10}$$

Therefore we get a map $\mathrm{Ad} : G \to \mathrm{GL}(\mathfrak{g})$. For matrix groups, this is given

$$\mathrm{Ad}_g(X) = gXg^{-1}, \qquad g \in G, \; X \in \mathfrak{g}, \tag{11}$$

and differentiating yields the Lie-algebra adjoint $\mathrm{ad}_X(Y) = [X, Y]$. One checks

$$\mathrm{Ad}_g([X, Y]) = [\mathrm{Ad}_g X, \mathrm{Ad}_g Y], \qquad \mathrm{ad}_X([Y, Z]) = [\mathrm{ad}_X Y, Z] + [Y, \mathrm{ad}_X Z]. \tag{12}$$

## A.3. Vectorized adjoint

Using the hat ($\wedge$) / vee ($\vee$) maps, the adjoint action on the Lie algebra induces a corresponding action on the vector coordinates. This vectorized action is a linear map represented by a matrix:

$$\mathrm{Ad}_g^m : \mathbb{R}^m \to \mathbb{R}^m, \qquad \mathrm{Ad}_g^m(x) = (\mathrm{Ad}_g(x^\wedge))^\vee. \tag{13}$$

Equivalently, $\mathrm{Ad}_g^m$ is the $m \times m$ matrix representation of $\mathrm{Ad}_g$ in the chosen basis $\{E_i\}$. In practice we precompute or assemble the $m \times m$ matrix representing $\mathrm{Ad}_g^m$ (or apply it implicitly) to implement left-multiplicative equivariant layers that act on vector features.

## A.4. Structure of Lie Algebra: Semisimplicity and Reductivity

**Definition A.1** (Semisimple and Reductive Lie Algebras)**.** A finite-dimensional Lie algebra $\mathfrak{g}$ is

- *semisimple* if it has no nonzero solvable ideals (equivalently, its radical is zero);

- *reductive* if it decomposes as a direct sum of ideals

$$\mathfrak{g} = \mathfrak{z}(\mathfrak{g}) \oplus [\mathfrak{g}, \mathfrak{g}], \tag{14}$$

where $\mathfrak{z}(\mathfrak{g})$ is the center and $[\mathfrak{g}, \mathfrak{g}]$ is semisimple.

**Example A.1.** *The Lie algebra $\mathfrak{gl}(n)$ decomposes as:*

$$\mathfrak{gl}(n) = \mathbb{R}I \oplus \mathfrak{sl}(n), \qquad \mathbb{R}I = \mathfrak{z}(\mathfrak{gl}(n)), \quad \mathfrak{sl}(n) = [\mathfrak{gl}(n), \mathfrak{gl}(n)]. \tag{15}$$

*where $\mathfrak{sl}(n)$ (traceless matrices) is semisimple, and $\mathbb{R}I$ (scalar matrices) forms the center. Thus $\mathfrak{gl}(n)$ is reductive but not semisimple (it has a nontrivial center).*

This decomposition highlights the non-semisimple nature of $\mathfrak{gl}(n)$, which plays a critical role in understanding the degeneracy of certain invariant forms such as the Killing form. This degeneracy hinders the application of standard tools in Lie-theoretic deep learning. Our work addresses this issue in the context of $\mathrm{GL}(n)$-equivariant architectures in Lie algebra $\mathfrak{gl}(n)$.

The Killing form on $\mathfrak{g}$ is

$$B_{\mathfrak{g}}(X, Y) := \mathrm{tr}(\mathrm{ad}_X \circ \mathrm{ad}_Y). \tag{16}$$

**Theorem A.2** (Cartan criterion). *For a finite-dimensional Lie algebra $\mathfrak{g}$, the Killing form $B_{\mathfrak{g}}$ is non-degenerate if and only if $\mathfrak{g}$ is semisimple.*

In particular, if $\mathfrak{g}$ has nontrivial center then $B_{\mathfrak{g}}$ is degenerate, since $\mathrm{ad}_Z = 0$ for all $Z \in \mathfrak{z}(\mathfrak{g})$.

### A.5. $\mathrm{Ad}$-invariant bilinear forms and the trace form

**Definition A.3** (Ad-invariance). A bilinear form $B : \mathfrak{g} \times \mathfrak{g} \to \mathbb{R}$ is $\mathrm{Ad}$-*invariant* if

$$B(\mathrm{Ad}_g X, \mathrm{Ad}_g Y) = B(X, Y) \qquad \forall g \in G, \ \forall X, Y \in \mathfrak{g}. \tag{17}$$

On semisimple $\mathfrak{g}$, the Killing form is $\mathrm{Ad}$-invariant and non-degenerate. On reductive but non-semisimple algebras (notably $\mathfrak{gl}(n)$), one often uses alternative $\mathrm{Ad}$-invariant forms. A basic example on $\mathfrak{gl}(n)$ is the trace pairing

$$\langle X, Y \rangle_{\mathrm{tr}} := \mathrm{tr}(XY). \tag{18}$$

**Proposition A.4** (Ad-invariance of the trace pairing on $\mathfrak{gl}(n)$). *For $g \in \mathrm{GL}(n)$ and $X, Y \in \mathfrak{gl}(n)$,*

$$\mathrm{tr}\big((gXg^{-1})(gYg^{-1})\big) = \mathrm{tr}(XY). \tag{19}$$

*Proof.* By cyclicity of trace, $\mathrm{tr}((gXg^{-1})(gYg^{-1})) = \mathrm{tr}(gXYg^{-1}) = \mathrm{tr}(XY)$. $\square$

The trace form is non-degenerate as a bilinear form on the vector space $\mathfrak{gl}(n)$ and therefore provides a practical substitute for the Killing form when designing $\mathrm{Ad}$-invariant bilinear layers on $\mathfrak{gl}(n)$.

We will use explicit $\mathfrak{gl}(n)$ identities (including the closed form of $B_{\mathfrak{gl}(n)}$, the resulting degeneracy, and our non-degenerate $\mathrm{Ad}$-invariant modification $\widetilde{B}$) in Appendix B.

## B. Connections for $\mathfrak{gl}(n)$ and $\mathfrak{so}(3) \simeq \mathbb{R}^3$

This appendix collects two specialization facts used in the main text: (i) an explicit expression for the (degenerate) Killing form on $\mathfrak{gl}(n)$ and its relation to our modified form $\widetilde{B}$; (ii) the standard intertwining between the $\mathrm{SO}(3)$ vector action and the adjoint action on $\mathfrak{so}(3)$.

**B.1. The classical Killing form on $\mathfrak{gl}(n)$**

**Proposition B.1** (Closed form and degeneracy). *For $X, Y \in \mathfrak{gl}(n)$,*

$$B_{\mathfrak{gl}(n)}(X, Y) = 2n \operatorname{tr}(XY) - 2 \operatorname{tr}(X) \operatorname{tr}(Y). \tag{20}$$

*In particular, $B_{\mathfrak{gl}(n)}$ is degenerate, and $\mathbb{R}I$ lies in its radical.*

*Proof sketch.* The identity (20) is standard for $\mathfrak{gl}(n)$ (up to an overall normalization convention). Degeneracy follows since $\operatorname{ad}_{\lambda I} = 0$ for all $\lambda I \in \mathbb{R}I$, hence $B_{\mathfrak{gl}(n)}(\lambda I, \cdot) = \operatorname{tr}(\operatorname{ad}_{\lambda I} \circ \operatorname{ad}.) = 0$. $\qquad \square$

**Proposition B.2** (Restriction to $\mathfrak{sl}(n)$). *If $X, Y \in \mathfrak{sl}(n)$, then*

$$B_{\mathfrak{gl}(n)}(X, Y) = 2n \operatorname{tr}(XY), \tag{21}$$

*which coincides with the Killing form on the semisimple ideal $\mathfrak{sl}(n)$ under the same normalization.*

*Proof.* For $X, Y \in \mathfrak{sl}(n)$, $\operatorname{tr}(X) = \operatorname{tr}(Y) = 0$ and (20) reduces to $2n \operatorname{tr}(XY)$. $\qquad \square$

**B.2. Our modified $\operatorname{Ad}$-invariant form on $\mathfrak{gl}(n)$**

In the main text we use

$$\widetilde{B}(X, Y) = 2n \operatorname{tr}(XY) - \operatorname{tr}(X) \operatorname{tr}(Y). \tag{22}$$

**Proposition B.3** (Relation to $B_{\mathfrak{gl}(n)}$). *With $B_{\mathfrak{gl}(n)}$ as in (20),*

$$\widetilde{B}(X, Y) = B_{\mathfrak{gl}(n)}(X, Y) + \operatorname{tr}(X) \operatorname{tr}(Y). \tag{23}$$

*Proof.* Subtract (20) from (22). $\qquad \square$

**Proposition B.4** (Orthogonal splitting). *Write $X = X_0 + \frac{1}{n} \operatorname{tr}(X)I$ and $Y = Y_0 + \frac{1}{n} \operatorname{tr}(Y)I$ with $X_0, Y_0 \in \mathfrak{sl}(n)$. Then*

$$\widetilde{B}(X, Y) = 2n \operatorname{tr}(X_0 Y_0) + \operatorname{tr}(X) \operatorname{tr}(Y), \tag{24}$$

*and $\mathbb{R}I$ and $\mathfrak{sl}(n)$ are $\widetilde{B}$-orthogonal.*

*Proof.* Using $\operatorname{tr}(I) = n$ and $\operatorname{tr}(X_0) = \operatorname{tr}(Y_0) = 0$,

$$\operatorname{tr}(XY) = \operatorname{tr}(X_0 Y_0) + \frac{1}{n} \operatorname{tr}(X) \operatorname{tr}(Y). \tag{25}$$

Substitute into (22). Orthogonality follows by setting $X = \lambda I$ and $Y \in \mathfrak{sl}(n)$. $\qquad \square$

**Proposition B.5** (Ad-invariance). *For all $g \in \operatorname{GL}(n)$ and $X, Y \in \mathfrak{gl}(n)$,*

$$\widetilde{B}(\operatorname{Ad}_g X, \operatorname{Ad}_g Y) = \widetilde{B}(X, Y). \tag{26}$$

*Proof.* Use $\operatorname{Ad}_g X = gXg^{-1}$, cyclicity $\operatorname{tr}(gXYg^{-1}) = \operatorname{tr}(XY)$, and $\operatorname{tr}(gXg^{-1}) = \operatorname{tr}(X)$ in (22). $\qquad \square$

**Proposition B.6** ($\widetilde{B}$ as an $\operatorname{Ad}$-invariant non-degenerate completion on the center). *The classical Killing form $B_{\mathfrak{gl}(n)}$ is degenerate with radical containing the center $\mathfrak{z}(\mathfrak{gl}(n)) = \mathbb{R}I$. The modified form $\widetilde{B}$ in (22) is $\operatorname{Ad}$-invariant and non-degenerate, and it agrees with $B_{\mathfrak{gl}(n)}$ on the semisimple ideal $\mathfrak{sl}(n)$ while supplying a non-degenerate inner product on the center:*

$$\widetilde{B}|_{\mathfrak{sl}(n) \times \mathfrak{sl}(n)} = B_{\mathfrak{sl}(n)}, \qquad \widetilde{B}(\lambda I, \mu I) = n^2 \lambda \mu \quad (\lambda, \mu \in \mathbb{R}). \tag{27}$$

*In particular, $\widetilde{B}$ resolves the degeneracy of $B_{\mathfrak{gl}(n)}$ precisely along $\mathbb{R}I$ without altering the semisimple part.*

*Proof.* Ad-invariance is Proposition B.5. For $\lambda I, \mu I \in \mathbb{R}I$, (22) gives $\widetilde{B}(\lambda I, \mu I) = 2n \operatorname{Tr}(\lambda \mu I) - \operatorname{Tr}(\lambda I) \operatorname{Tr}(\mu I) = 2n(\lambda \mu n) - (\lambda n)(\mu n) = n^2 \lambda \mu$, which is non-degenerate. The restriction to $\mathfrak{sl}(n)$ follows since $\operatorname{Tr}(X) = 0$ on $\mathfrak{sl}(n)$, so (22) reduces to $2n \operatorname{Tr}(XY)$, which matches Proposition B.2 under the same normalization. $\qquad \square$

**B.3. $\mathfrak{so}(3) \simeq \mathbb{R}^3$: vector action vs. adjoint action**

Let $\hat{\cdot} : \mathbb{R}^3 \to \mathfrak{so}(3)$ be the standard hat map defined by $\hat{v} w = v \times w$, with inverse $(\cdot)^\vee : \mathfrak{so}(3) \to \mathbb{R}^3$.

**Lemma B.7** (Intertwining identity). *For all $R \in \mathrm{SO}(3)$ and $v \in \mathbb{R}^3$,*

$$\widehat{Rv} = R\hat{v}R^\top = \mathrm{Ad}_R(\hat{v}), \qquad \mathrm{Ad}_R(X) = RXR^{-1} = RXR^\top. \tag{28}$$

*Consequently, if $F : \mathfrak{so}(3) \to \mathfrak{so}(3)$ is $\mathrm{Ad}$-equivariant, then $f(v) := \big(F(\hat{v})\big)^\vee$ is left $\mathrm{SO}(3)$-equivariant: $f(Rv) = Rf(v)$.*

*Proof.* For any $w \in \mathbb{R}^3$, using $(Ra) \times (Rb) = R(a \times b)$,

$$(R\hat{v}R^\top)w = R\hat{v}(R^\top w) = R\big(v \times (R^\top w)\big) = (Rv) \times w = \widehat{Rv}\, w. \tag{29}$$

Thus $\widehat{Rv} = R\hat{v}R^\top$. The equivariance statement follows by applying $(\cdot)^\vee$ and using $\widehat{Ra} = R\hat{a}R^\top$. $\qquad\square$

*Remark* B.8 (Trace contraction recovers the Vector Neuron inner product). Under the hat identification $\hat{\cdot} : \mathbb{R}^3 \to \mathfrak{so}(3)$,

$$\hat{v} = \begin{bmatrix} 0 & -v_3 & v_2 \\ v_3 & 0 & -v_1 \\ -v_2 & v_1 & 0 \end{bmatrix}, \qquad \mathrm{Tr}(\hat{v}\,\hat{w}) = -2\,v^\top w \quad (v, w \in \mathbb{R}^3). \tag{30}$$

Hence the $\mathrm{Ad}$-invariant trace pairing on $\mathfrak{so}(3)$, $\langle X, Y \rangle_{\mathrm{tr}} := -\frac{1}{2}\mathrm{Tr}(XY)$, corresponds exactly to the Euclidean dot product on $\mathbb{R}^3$: $\langle \hat{v}, \hat{w} \rangle_{\mathrm{tr}} = v^\top w$. In particular, $\mathrm{Ad}$-invariant scalar contractions on $\mathfrak{so}(3)$ recover the invariants used by Vector Neurons (Deng et al., 2021)(up to a fixed global scaling, which can be absorbed by adjacent learnable linear maps or normalization).

# C. Proofs of Key Theorems

In this section, we provide a generalized treatment and proofs for the modified bilinear form $\widetilde{B}$ introduced in Section 4 and Appendix B. Our discussion here establishes the formal properties of $\widetilde{B}$ for any real reductive Lie algebra.

## C.1. Proof of Non-degeneracy and Ad-invariance of Modified Killing Form $B_e$

Let $\mathfrak{g}$ be a real reductive Lie algebra.

**Definition C.1** (Reductive decomposition). A Lie algebra $\mathfrak{g}$ is *reductive* if $\mathfrak{g} = \mathfrak{z}(\mathfrak{g}) \oplus [\mathfrak{g}, \mathfrak{g}]$, where $\mathfrak{z}(\mathfrak{g})$ is the center and $[\mathfrak{g}, \mathfrak{g}]$ is semisimple. This decomposition is canonical (both summands are ideals); see Kirillov (2008) for a standard treatment.

**Definition C.2** (Modified Killing form on a reductive Lie algebra). Fix any symmetric, positive–definite inner product $\langle \cdot, \cdot \rangle_{\mathfrak{z}}$ on $\mathfrak{z}(\mathfrak{g})$, and let $B$ denote the Killing form on the semisimple ideal $[\mathfrak{g}, \mathfrak{g}]$. For $Z_i \in \mathfrak{z}(\mathfrak{g})$ and $X_i \in [\mathfrak{g}, \mathfrak{g}]$ define

$$\widetilde{B}(Z_1 + X_1, \, Z_2 + X_2) := \langle Z_1, Z_2 \rangle_{\mathfrak{z}} + B(X_1, X_2). \tag{31}$$

*Remark* C.3 (Canonicity). On $[\mathfrak{g}, \mathfrak{g}]$ the restriction (Killing form) is canonical. On $\mathfrak{z}(\mathfrak{g})$ there is no canonical choice; any $\mathrm{Ad}$-invariant positive-definite inner product works. The choice we make in the case of $\mathfrak{gl}(n)$ ensures that it agrees with the Killing form on the semisimple part $\mathfrak{sl}(n)$, and the center $\mathbb{R}I$ is normalized by a natural trace scale.

**Proposition C.4** (Block–orthogonality and restrictions). *With notation as above,*

$$\widetilde{B}\big(\mathfrak{z}(\mathfrak{g}), [\mathfrak{g}, \mathfrak{g}]\big) = 0, \qquad \widetilde{B}\big|_{\mathfrak{z}(\mathfrak{g})} = \langle \cdot, \cdot \rangle_{\mathfrak{z}}, \qquad \widetilde{B}\big|_{[\mathfrak{g}, \mathfrak{g}]} = B. \tag{32}$$

**Proposition C.5** (Non–degeneracy). $\widetilde{B}$ *is nondegenerate on $\mathfrak{g}$.*

*Proof.* Let $X = Z + W$ with $Z \in \mathfrak{z}(\mathfrak{g})$ and $W \in [\mathfrak{g}, \mathfrak{g}]$. If $\widetilde{B}(X, \cdot) \equiv 0$, then testing against $Y \in \mathfrak{z}(\mathfrak{g})$ yields $\langle Z, Y \rangle_{\mathfrak{z}} = 0$ for all $Y$, hence $Z = 0$; testing against $Y \in [\mathfrak{g}, \mathfrak{g}]$ yields $B(W, Y) = 0$ for all $Y$, hence $W = 0$ by the non–degeneracy of $B$ on the semisimple ideal. Thus $X = 0$. $\qquad\square$

**Proposition C.6** (Ad–invariance on the identity component). $\widetilde{B}$ *is* ad–*invariant:*

$$\widetilde{B}([X,Y],Z) + \widetilde{B}(Y,[X,Z]) = 0 \qquad \textit{for all } X, Y, Z \in \mathfrak{g}, \tag{33}$$

*and hence* $\widetilde{B}(\mathrm{Ad}_g Y, \mathrm{Ad}_g Z) = \widetilde{B}(Y, Z)$ *for all $g$ in the identity component $G^\circ$.*

*Proof.* The restriction to $[\mathfrak{g}, \mathfrak{g}]$ equals $B$, which is ad–invariant. If $Z \in \mathfrak{z}(\mathfrak{g})$ then $[X, Z] = 0$ for all $X$, so any bilinear form on $\mathfrak{z}(\mathfrak{g})$ is automatically ad–invariant. Using Proposition C.4 and bilinearity gives the displayed identity. Equivalence with Ad–invariance on $G^\circ$ follows by integrating the infinitesimal relation along paths in $G^\circ$. $\qquad \square$

*Remark* C.7 (Invariance for nonconnected groups). In case the group is nonconnected, and one desires invariance under the full group $G$ (not just $G^\circ$). The component group $\Gamma = G/G^\circ$ acts linearly on $\mathfrak{z}(\mathfrak{g})$. In all practical cases, $\Gamma$ will be a finite group. Then averaging any positive–definite $\langle \cdot, \cdot \rangle_\mathfrak{z}$ over $\Gamma$ yields an $\mathrm{Ad}(G)$–invariant inner product on the center:

$$\langle Z_1, Z_2 \rangle_\mathfrak{z}^{\mathrm{avg}} = \frac{1}{|\Gamma|} \sum_{\gamma \in \Gamma} \left\langle \mathrm{Ad}_\gamma Z_1, \, \mathrm{Ad}_\gamma Z_2 \right\rangle_\mathfrak{z}. \tag{34}$$

Replacing $\langle \cdot, \cdot \rangle_\mathfrak{z}$ by $\langle \cdot, \cdot \rangle_\mathfrak{z}^{\mathrm{avg}}$ in Equation 31 makes $\widetilde{B}$ invariant under all of $G$.

## D. Summary of Key Lie Groups and Algebras

To provide a comprehensive overview of the geometric structures discussed in this work, Table 8 summarizes the Lie groups $G$, their associated Lie algebras $\mathfrak{g}$, and their application domains. We categorize each group by its algebraic properties—specifically distinguishing between reductive, semisimple, and compact types—and detail their specific roles within both the broader literature and our experimental framework. This unified taxonomy situates ReLNs as a general-purpose backbone capable of handling a wide array of linear symmetries encountered in scientific machine learning.

*Table 8.* Comprehensive survey of Lie groups, algebras, and their applications. We highlight the algebraic classifications that dictate the choice of invariant bilinear forms and contrast general literature examples with our specific experimental tasks.

| Group ($G$) | Algebra ($\mathfrak{g}$) | Algebraic Type | General Applications & Refs | Relevance to ReLN Framework |
| --- | --- | --- | --- | --- |
| **General Linear** | $\mathfrak{gl}(n)$ | Reductive | General linear transformations, stress-strain tensors (Basu et al., 2025; Finzi et al., 2021). | **Core Domain.** Primary target of ReLNs; resolves the Killing form degeneracy on $\mathfrak{gl}(n)$. |
| **Special Linear** | $\mathfrak{sl}(3)$ | Semisimple | Homography classification, 3D vision, SLAM (Lin et al., 2024a; Finzi et al., 2021). | **Algebraic Benchmark.** Validates generalization on semisimple non-compact subgroups (Section 5.2.1). |
| **Symplectic** | $\mathfrak{sp}(4)$ | Semisimple | Hamiltonian mechanics, phase space dynamics (Lin et al., 2024a; Finzi et al., 2021). | **Algebraic Benchmark.** Tests the ability to learn highly nonlinear invariants on conserved systems (Section 5.2.2). |
| **Special Orthogonal** | $\mathfrak{so}(3)$ | Compact | 3D Point clouds, state estimation, rigid body rotations (Deng et al., 2021; Son et al., 2024; Thomas et al., 2018). | **Robotics Application.** Lifts 3D velocity vectors into $\mathfrak{so}(3) \subset \mathfrak{gl}(3)$ (Sections 5.3 and 5.4). |
| **Lorentz Group** | $\mathfrak{so}(1,3)^+$ | Semisimple | Particle physics, jet tagging, relativistic collisions (Bogatskiy et al., 2020; Batatia et al., 2023). | **Physics Application.** 4-momenta embedded into $\mathfrak{gl}(5)$ to unify left-action with adjoint action (Section I.1. |
| **Special Unitary** | $\mathfrak{su}(3)$ | Compact | Quantum Chromodynamics (QCD), lattice gauge theory (Favoni et al., 2022). | **General Example.** Illustrates framework applicability to complex-valued compact groups. |
| **SPD Manifold**[†] | $\mathrm{SPD}(n)$ | Riemannian | Geometric uncertainty, inertia tensors, diffusion MRI (Ma et al., 2025; Magnus, 1985). | **Geometric Uncertainty.** Covariances mapped to $\mathfrak{gl}(n)$ (Sections 5.3 and 5.4). |

[†] Note: $\mathrm{SPD}(n)$ is not a group but a manifold representable as the quotient space $\mathrm{GL}(n)/\mathrm{O}(n)$. We process it by mapping to the linear subspace $\mathfrak{sym}(n) \subset \mathfrak{gl}(n)$ via the matrix logarithm (Ma et al., 2025).

## E. Detailed Layer Formulations

This section provides the precise mathematical definitions and equivariance proofs for the ReLN architecture. To maintain consistency with the main text, we denote the input tensor as $x \in \mathbb{R}^{B \times K \times C}$, where $B$ is the batch size, $K = \dim \mathfrak{g}$ is the geometric dimension, and $C$ is the number of feature channels. Each column $x_{b,:,c} \in \mathbb{R}^K$ corresponds to a matrix $X_{b,c} \in \mathfrak{g}$ via the vee/hat isomorphism.

### E.1. Equivariant Linear Layer

The ReLN-Linear layer applies a linear map to the channel dimension (the last axis). For an input $x \in \mathbb{R}^{B \times K \times C}$ and weights $W \in \mathbb{R}^{C \times C'}$, the operation is defined as:

$$f_{\text{ReLN-Lin}}(x; W) = xW \in \mathbb{R}^{B \times K \times C'}. \tag{35}$$

We omit any bias term to preserve exact equivariance.

**Proof of Equivariance.** The group action, $\text{Ad}_g$ (defined in Equation 11), is a linear map that multiplies each feature channel from the left. The weight matrix $W$ multiplies the channel dimension from the right. These operations commute, ensuring strict $G$-equivariance for any $g \in G$:

$$\begin{aligned} f_{\text{ReLN-Lin}}(\text{Ad}_g(x); W) &= (\text{Ad}_g x)W \\ &= \text{Ad}_g(xW) \\ &= \text{Ad}_g(f_{\text{ReLN-Lin}}(x; W)). \end{aligned} \tag{36}$$

### E.2. Equivariant Nonlinearities

Standard pointwise activations break equivariance under non-orthogonal transforms. We introduce two equivariant alternatives.

**ReLN-ReLU.** This layer rectifies a feature based on its alignment with a learnable direction. Given the input $x \in \mathbb{R}^{B \times K \times C}$, we first compute equivariant directions $d = f_{\text{ReLN-Lin}}(x; U)$ where $U \in \mathbb{R}^{C \times C}$. The nonlinearity is then applied channel-wise as a scalar-gated update:

$$f_{\text{ReLN-ReLU}}(x)_{b,:,c} = \begin{cases} x_{b,:,c}, & \text{if } \widetilde{B}(x_{b,:,c}^\wedge, d_{b,:,c}^\wedge) \leq 0, \\ x_{b,:,c} + \sigma\big(\widetilde{B}(x_{b,:,c}^\wedge, d_{b,:,c}^\wedge)\big)d_{b,:,c}, & \text{otherwise}, \end{cases} \tag{37}$$

where $\sigma(t) = \max(0, t)$ denotes the standard ReLU activation function acting as an invariant scalar gate. Since the modified bilinear form $\widetilde{B}$ is $\text{Ad}$-invariant, the gate $\sigma(\widetilde{B}(\cdot, \cdot))$ is group-invariant. Furthermore, because $d$ transforms equivariantly with $x$, the resulting vector update preserves strict $\text{Ad}_g$-equivariance. The leaky variant $f_{\text{ReLN-LeakyReLU}}(x) = \alpha x + (1 - \alpha)f_{\text{ReLN-ReLU}}(x)$ follows directly.

**ReLN-Bracket (Lie-bracket nonlinearity).** This layer uses the matrix commutator, a natural $\text{Ad}$-equivariant primitive, to create learnable interactions between channels. Let the input be a batch of features represented by their vector coordinates, $x \in \mathbb{R}^{B \times K \times C_{\text{in}}}$.

First, two independent linear maps with weights $W_a, W_b \in \mathbb{R}^{C_{\text{in}} \times C_{\text{out}}}$ produce intermediate tensors $u = xW_a$ and $v = xW_b$. The Lie bracket is then computed channel-wise between the corresponding feature vectors of $u$ and $v$ to produce an update tensor $\Delta x \in \mathbb{R}^{B \times K \times C_{\text{out}}}$:

$$(\Delta x)_{b,:,c'} = \left[(u_{b,:,c'})^\wedge, (v_{b,:,c'})^\wedge\right]^\vee. \tag{38}$$

This update is added to the input via a residual connection (requiring $C_{\text{in}} = C_{\text{out}}$):

$$f_{\text{ReLN-Bracket}}(x) = x + \Delta x. \tag{39}$$

Each step in this process is equivariant under the adjoint action, making the entire block exactly $\text{Ad}_g$-equivariant: $f_{\text{ReLN-Bracket}}(\text{Ad}_g x) = \text{Ad}_g(f_{\text{ReLN-Bracket}}(x))$ for all $g \in G$.

## F. Theoretical Complexity and Scaling Analysis

To address the operational properties of our architecture, we provide an explicit Big-$\mathcal{O}$ complexity analysis connecting the feature channel width $C$ and the Lie algebra matrix dimension $n$. We further validate this analysis with empirical benchmarks.

**1. Theoretical Scaling with Matrix Dimension $n$.** We fix the **total number of scalars per token** to $C$ to ensure a fair comparison. The effective channel width $M$ is scaled such that $M \times K = C$.

- **Standard MLP ($K = 1$):** Uses full width $M = C$.

- **ReLN ($K = n^2$):** Operates on $\mathfrak{gl}(n)$ matrices. Effective width $M = C/n^2$.

Let $n$ be the dimension of the matrices (e.g., $n = 3$ for $\mathfrak{gl}(3)$).

- **Linear Layer:** The layer performs $K$ independent matrix multiplications of size $M \times M$.

$$\text{Cost}_{\text{linear}} = K \cdot M^2 = n^2 \cdot \left(\frac{C}{n^2}\right)^2 = \frac{C^2}{n^2}. \tag{40}$$

  The dominant quadratic cost **decreases** as the matrix dimension $n$ increases. For $n = 3$, ReLN reduces the Linear Layer FLOPs by a factor of $1/9$ compared to an MLP ($n = 1$).

- **Lie Bracket Nonlinearity:** This operation computes $[X, Y] = XY - YX$, involving $n \times n$ matrix multiplications.

$$\text{Cost}_{\text{bracket}} = M \cdot \mathcal{O}(n^3) = \frac{C}{n^2} \cdot \mathcal{O}(n^3) = \mathcal{O}(C \cdot n). \tag{41}$$

  In deep learning contexts where $C$ is large (e.g., $C \geq 128$), the quadratic term $\mathcal{O}(C^2/n^2)$ dominates the linear term $\mathcal{O}(Cn)$.

**2. Layer-wise Complexity Comparison.** Table 9 presents the theoretical breakdown. ReLN-Bracket reduces the total block cost to $\approx 12.5\%$ of the baseline.

*Table 9.* **Theoretical Complexity Analysis.** Comparison under a fixed feature budget $C$. By utilizing a higher-dimensional geometric space ($n = 3, K = 9$), ReLN reduces the effective channel width to $C/9$, resulting in a quadratic reduction in the dominant Linear Layer cost.

| Layer Type | Model | Eff. Channels ($M$) | Parameters | FLOPs | Relative Cost |
|---|---|---|---|---|---|
| **Linear Layer** | MLP ($n = 1$) | $C$ | $C^2$ | $C^2$ | $1.00\times$ |
| | VN ($n \approx 1.7$) | $C/3$ | $0.33C^2$ | $0.33C^2$ | $0.33\times$ |
| | **ReLN ($n = 3$)** | **$C/9$** | **$0.11C^2$** | **$0.11C^2$** | **$0.11\times$** |
| **Nonlinearity** | ReLU (MLP) | $C$ | $0$ | $\mathcal{O}(C)$ | $\approx 0.00\times$ |
| | VN-ReLU | $C/3$ | $0.11C^2$ | $0.11C^2$ | $0.11\times$ |
| | ReLN-ReLU | $C/9$ | $0.01C^2$ | $0.01C^2$ | $0.01\times$ |
| | **ReLN-Bracket** | **$C/9$** | **$0.025C^2$** | **$0.025C^2$** | **$0.03\times$** |

# G. Experimental Details

Training and evaluation for all presented experiments, Platonic solid classification, invariant function regression, top tagging, and drone state estimation, were conducted on a single NVIDIA GeForce RTX 4090 GPU.

## G.1. Model Architectures and Implementation Details

Across all experiments, our proposed ReLN models are constructed by stacking ReLN-Linear, ReLN-ReLU, and ReLN-Bracket layers. The specific number of layers and channel widths are adapted for each task to ensure a fair comparison with baseline models in terms of parameter count.

**Algebraic Benchmarks ($\mathfrak{sl}(3)$ and $\mathfrak{sp}(4)$).** For the Platonic solid classification and $\mathfrak{sp}(4)$ invariant regression tasks, our ReLN model directly adopts the architecture used by the Lie Neurons benchmark model from Lin et al. (2024a). The primary modification is the replacement of their Killing form-based nonlinearity and invariant layers with our proposed non-degenerate bilinear form $B$ (Eq. 2). This setup allows for a direct comparison of the impact of the bilinear form, as all other architectural hyperparameters are kept identical to the baseline.

**Top Tagging.** For the Top-Tagging task, our model is a modification of the LorentzNet architecture (Gong et al., 2022). We adapt its Lorentz Group Equivariant Blocks (LGEBs) by replacing the invariant feature computation with our proposed bilinear form. A detailed description of the architecture, our modifications, and training protocol is provided in Appendix I.

**Drone State Estimation.** In this task, we compare our ReLN model against two baseline families: a non-equivariant 1D ResNet and an equivariant Vector Neurons (VN) model. The specific implementation details and architectural choices for each model are provided next in Appendix J.

### G.2. Platonic Solid Classification on $\mathfrak{sl}(3)$

**Overview.** All experiments evaluate classification of Platonic solids (tetrahedron, octahedron, icosahedron) from inter-face homographies computed in the image plane. For each model-family we train 5 independent runs with different random seeds and report mean ± standard deviation. Training uses fixed object and camera poses; at test time we report results on the in-distribution (ID) split and the rotated-camera (RC) split (RC applies ten random $\mathrm{SO}(3)$ rotations to the camera frame). The 'MLP (wider)' denotes a capacity-matched ($\approx 2\times$ parameters) MLP used for a fairer comparison.

*Table 10.* Common training hyperparameters (used across model families unless noted).

| Hyperparameter | Value |
|---|---|
| Optimizer | Adam |
| Batch size | 100 |
| Number of independent runs (seeds) | 5 |
| Max epochs / stopping criterion | 5000 epochs |
| Data augmentation (train) | Random camera rotations applied to training examples when enabled |
| RC evaluation | 500 random $\mathrm{SO}(3)$ rotations applied to each test example |
| Metric reported | Classification accuracy (mean ± std across runs) |

*Table 11.* Model-specific hyperparameters and implementation notes.

| Model family | Key choices | Notes |
|---|---|---|
| Latent Feature Size (MLP Baseline) | 256 | As in Lin et al. (2024a). |
| Latent Feature Size (MLP Wider) | 386 | Increased width total parameters $\approx 2\times$ baseline. |
| Learning rate (MLP models) | $1 \times 10^{-4}$ | |
| Learning rate (ReLNs/Lie Neurons models) | $3 \times 10^{-6}$ | Lower LR chosen for stable training |

## H. System Identification: Experimental Details

This appendix details the experimental protocol for the native $\mathrm{GL}(n)$ system-identification task of Section 5.1, including the precise definitions of the evaluation metrics, the paired data-generation procedure, and the full per-split results. The final subsection reports a center-scale ablation that supports the choice of $\mathrm{Ad}$-invariant inner product on the center.

### H.1. Task and Metrics

We consider discrete-time linear dynamics $z_{t+1} = A z_t + w_t$ with state $z_t \in \mathbb{R}^3$, unknown transition matrix $A \in \mathfrak{gl}(3)$, and process noise $w_t \sim \mathcal{N}(0, \sigma^2 I)$. For each trajectory $(z_1, \ldots, z_{L+1})$ of length $L$, the windowed least-squares estimator

$$\widetilde{A} = \arg \min_{B \in \mathbb{R}^{3 \times 3}} \sum_{t=1}^{L} \|z_{t+1} - B z_t\|^2 \tag{42}$$

yields a noisy point estimate of $A$. The task input is a set of $N$ such estimates $\{\widetilde{A}_i\}_{i=1}^{N}$ from independently sampled trajectories of the same system; the target is the global transition matrix $A$.

**Similarity transforms.** Under a basis change $z \mapsto Sz$ with $S \in \mathrm{GL}(3)$, both inputs and target transform by similarity:

$$\widetilde{A}_i \mapsto S \widetilde{A}_i S^{-1}, \qquad A \mapsto S A S^{-1}. \tag{43}$$

This is the adjoint action on $\mathfrak{gl}(3)$. We evaluate three test conditions, all using the same underlying systems with different basis-change distributions:

- **ID:** $S = I$ (training basis).

- **GL:** $S = U\Lambda V^\top$ with $U, V$ drawn from the Haar measure on O(3) and $\Lambda = \mathrm{diag}(\kappa, \lambda_2, 1)$, $\kappa \in [2, 10]$, $\lambda_2 \in [1, \kappa]$.

- **GL-Hard:** Same construction as GL but with $\kappa \in [10, 40]$.

The condition number ranges control the difficulty of the basis change; larger $\kappa$ pushes $S$ further from the orthogonal subgroup.

**Metrics.** For a prediction $\hat{A}$ produced in the transformed basis $S$, we report two metrics computed against the corresponding ground truth:

$$\textbf{Trace MSE} \;=\; \mathbb{E}\big[(\mathrm{Tr}\,\hat{A} - \mathrm{Tr}\,A_{\text{target}})^2\big], \tag{44}$$

$$\textbf{Canonical MSE} \;=\; \mathbb{E}\big[\|S^{-1}\hat{A}S - A\|_F^2\big], \tag{45}$$

where $A$ is the system in the canonical (pre-transform) basis. The *Trace MSE* measures recovery of the central component of $A$ and is invariant under similarity ($\mathrm{Tr}(SAS^{-1}) = \mathrm{Tr}\,A$); a model that cannot represent the center will fail on this metric even at $S = I$. The *Canonical MSE* maps the prediction back to the canonical frame before computing the residual, so a similarity-equivariant model is rewarded for representing $A$ up to the test-time basis change.

## H.2. Data Generation

We adopt a *paired* design in which the same set of underlying systems is tested under all three transforms, isolating the effect of $S$.

**Canonical bank.** For each system we draw a base $A_0 \in \mathfrak{sl}(3)$ from a Gaussian ensemble (i.i.d. $\mathcal{N}(0,1)$ entries, trace removed) and a center coefficient $\alpha \sim \mathrm{Unif}[-2.5, 2.5]$, forming $A = A_0 + \alpha I$; the wide center range ensures that the trace carries non-trivial signal. To prevent unstable rollouts, $A$ is normalized to unit spectral norm via $A \leftarrow A/\sigma_{\max}(A)$. From each system we generate $N = 5$ trajectories of length $L = 20$ with i.i.d. noise $w_t \sim \mathcal{N}(0, \sigma^2 I)$, $\sigma = 0.1$, and compute one least-squares estimate per trajectory, yielding $\{\widetilde{A}_i\}_{i=1}^5$. The training bank contains 2,000 systems and the test bank 500.

**Transform application.** The transform $S$ is constructed as $S = U\Lambda V^\top$ with $U, V$ obtained as the orthogonal factors of QR decompositions of standard Gaussian matrices (Haar-distributed on O(3)) and $\Lambda$ as described in Section H.1. We then form similarity-transformed inputs $S\widetilde{A}_i S^{-1}$ and targets $SAS^{-1}$. The training set uses $S = I$ exclusively; test-time basis changes are applied only at evaluation, so models do not see any transformed example during training.

## H.3. Architectures, Training, and Full Results

**ReLN and LN.** Both equivariant models share a point-wise encoder–pooling–decoder structure operating on the set $\{\widetilde{A}_i\}$. Each block consists of a ReLN-Linear map, an $\mathrm{Ad}$-equivariant batch normalization, a $\widetilde{B}$-gated ReLU (ReLN-ReLU), and a Lie-bracket nonlinearity in the second block. Hidden channel width is $C = 16$ and mean pooling is applied across the 5 windowed estimates. ReLN operates on the full $\mathfrak{gl}(3)$ representation; LN operates on $\mathfrak{sl}(3)$ with the input trace explicitly removed before the network, matching its semisimple construction.

**Baselines.** The MLP baseline flattens each $3 \times 3$ estimate to a 9-dimensional vector and applies two GELU-MLP blocks with hidden width 64, followed by mean pooling and a linear decoder. The structured *Avg-LS* baseline returns the unweighted average $\frac{1}{N}\sum_i \widetilde{A}_i$, providing a parameter-free reference that is exactly similarity-equivariant.

**Training.** All models are trained with Adam ($\mathrm{lr} = 10^{-3}$), batch size 128, for 50 epochs, minimizing the Frobenius loss $\|\hat{A} - A_{\text{target}}\|_F^2$ on the training set ($S = I$). Each configuration is run with 3 seeds $\{42, 2026, 777\}$.

The equivariant models (Avg-LS, Lie Neurons, ReLN) yield identical metrics across the three splits, confirming exact similarity equivariance and matching the proof of Section 4. The MLP shows the characteristic non-equivariant failure mode: Canonical MSE grows by four orders of magnitude between ID and GL-Hard. Lie Neurons attains a Trace MSE two orders of magnitude higher than ReLN, since Killing-form-only invariants on $\mathfrak{sl}(3)$ are unable to represent $\mathrm{Tr}\,A$ by construction.

*Table 12.* Full System Identification results (mean $\pm$ std over 3 seeds). The MLP collapses on shifted bases; Lie Neurons remain stable in Canonical MSE but cannot recover the trace; ReLN attains the lowest error on both metrics across all splits. The three equivariant methods (Avg-LS, Lie Neurons, ReLN) produce identical metrics across the three test conditions, confirming exact similarity equivariance.

| Method | Split | Trace MSE $\downarrow$ | Canonical MSE $\downarrow$ |
|---|---|---|---|
| Avg-LS | ID | $0.0211 \pm 0.0004$ | $0.0044 \pm 0.0002$ |
| Avg-LS | GL | $0.0211 \pm 0.0004$ | $0.0044 \pm 0.0002$ |
| Avg-LS | GL-Hard | $0.0211 \pm 0.0004$ | $0.0044 \pm 0.0002$ |
| MLP | ID | $0.0145 \pm 0.0005$ | $0.0041 \pm 0.0002$ |
| MLP | GL | $0.0489 \pm 0.0064$ | $0.1480 \pm 0.0220$ |
| MLP | GL-Hard | $1.0285 \pm 0.0270$ | $57.4099 \pm 7.3559$ |
| Lie Neurons | ID | $1.6850 \pm 0.0500$ | $0.0658 \pm 0.0017$ |
| Lie Neurons | GL | $1.6850 \pm 0.0500$ | $0.0658 \pm 0.0017$ |
| Lie Neurons | GL-Hard | $1.6850 \pm 0.0500$ | $0.0658 \pm 0.0017$ |
| **ReLN** | **ID** | $\mathbf{0.0133 \pm 0.0005}$ | $\mathbf{0.0039 \pm 0.0002}$ |
| **ReLN** | **GL** | $\mathbf{0.0133 \pm 0.0005}$ | $\mathbf{0.0039 \pm 0.0002}$ |
| **ReLN** | **GL-Hard** | $\mathbf{0.0133 \pm 0.0005}$ | $\mathbf{0.0039 \pm 0.0002}$ |

*Table 13.* Center-scale ablation. Varying $\alpha$ over two orders of magnitude leaves performance unchanged within seed-level variance, confirming that the choice of $\mathrm{Ad}$-invariant inner product on a one-dimensional center is absorbed by training.

| Center scale $\alpha$ | Trace MSE $\downarrow$ | Canonical MSE $\downarrow$ |
|---|---|---|
| 0.1 | $0.0135 \pm 0.0002$ | $0.0040 \pm 0.0002$ |
| 0.5 | $0.0127 \pm 0.0006$ | $0.0038 \pm 0.0002$ |
| 1.0 | $0.0133 \pm 0.0005$ | $0.0039 \pm 0.0002$ |
| 2.0 | $0.0130 \pm 0.0004$ | $0.0039 \pm 0.0002$ |
| 10.0 | $0.0140 \pm 0.0004$ | $0.0041 \pm 0.0002$ |

## H.4. Center-Scale Ablation

The completion in Definition 4.1 admits any $\mathrm{Ad}$-invariant inner product on the center; for $\mathfrak{gl}(n)$ the center is one-dimensional, so all such choices reduce to a single positive scaling. We verify empirically that this scaling is absorbed by adjacent learnable maps and does not affect performance.

We parameterize $\widetilde{B}_\alpha(X, Y) = 2n \operatorname{Tr}(X_0 Y_0) + \alpha \operatorname{Tr}(X) \operatorname{Tr}(Y), \quad X_0 = X - \frac{1}{n} \operatorname{Tr}(X) I$ and sweep $\alpha$ over two orders of magnitude on the System Identification task, with all other hyperparameters fixed.

For reductive Lie algebras with higher-dimensional centers, this equivalence no longer holds: non-proportional $\mathrm{Ad}$-invariant inner products on $\mathfrak{z}(\mathfrak{g})$ can induce distinct inductive biases. A principled choice for such cases lies outside the scope of this work.

# I. Top Tagging Experiment: Framework, Proof, and Implementation

This appendix provides details for our Lorentz-equivariant jet tagging study. We first summarize the task and results, then present the Lorentz-compatible embedding and its equivariance proof, and finally describe the model and training protocol.

## I.1. Particle Physics with Lorentz Group $\mathrm{SO}^+(1, 3)$ Equivariance

We evaluate on the Top-Tagging benchmark (Kasieczka et al., 2019), which classifies particle jets originating from top quarks against a Quantum Chromodynamics background. Since relativistic collisions respect spacetime symmetries, the model should be equivariant under the proper orthochronous Lorentz group $\mathrm{SO}^+(1, 3)$. To express this as an adjoint action within our Lie-algebraic framework, we embed each four-momentum $p \in \mathbb{R}^{1,3}$ into $\mathfrak{gl}(5)$ via

$$\varphi(p) = \begin{bmatrix} 0_{4 \times 4} & p \\ p^\top \eta & 0 \end{bmatrix}, \qquad \eta = \operatorname{diag}(-1, 1, 1, 1), \tag{46}$$

and show in Appendix I.2 that $\varphi(\Lambda p) = \mathrm{Ad}_{\operatorname{diag}(\Lambda, 1)}\big(\varphi(p)\big)$ for $\Lambda \in \mathrm{SO}^+(1, 3)$.

For comparison, we modify LorentzNet by replacing its invariant feature computation with our bilinear-form construction

*Table 14.* Comparison of performance on the Top-Tagging dataset. ReLN results include both the parameter-efficient version (84k) and the parameter-matched version (224k). Rej@30% denotes background rejection at 30% signal efficiency (higher is better). Benchmark scores are as reported in the original publications.

| Architecture | #Params | Acc. | AUC | Rej@30% | Reference |
|---|---|---|---|---|---|
| PELICAN | 45k | 0.943 | 0.987 | $2289 \pm 204$ | Bogatskiy et al. (2022) |
| LorentzNet (orig.) | 224k | 0.942 | 0.987 | $2195 \pm 173$ | Gong et al. (2022) |
| LorentzNet (reprod.) | 84k | 0.942 | 0.987 | $1821 \pm 94$ | Our reprod. |
| LGN | 4.5k | 0.929 | 0.964 | $435 \pm 95$ | Bogatskiy et al. (2020) |
| BIP | 4k | 0.931 | 0.981 | $853 \pm 68$ | Munoz et al. (2022) |
| partT | 2.14M | 0.940 | 0.986 | $1602 \pm 81$ | Qu et al. (2022) |
| ParticleNet | 498k | 0.938 | 0.985 | $1298 \pm 46$ | Qu & Gouskos (2020) |
| EFN | 82k | 0.927 | 0.979 | $633 \pm 31$ | Komiske et al. (2019) |
| TopoDNN | 59k | 0.916 | 0.972 | $295 \pm 5$ | Pearkes et al. (2017) |
| LorentzMACE | 228k | 0.942 | 0.987 | $1935 \pm 85$ | Batatia et al. (2023) |
| CGENN | 321K | 0.942 | 0.987 | 2172 | Ruhe et al. (2023) |
| **Lie Neurons** | 224k | 0.941 | 0.985 | $1655 \pm 73$ | Our reprod. |
| **ReLN (Ours)** | 224k | 0.942 | 0.987 | $2201 \pm 101$ | |
| **ReLN (Ours)** | 84k | 0.942 | 0.987 | $1979 \pm 87$ | |

and additionally report a parameter-matched LorentzNet baseline. Table 14 shows that ReLN achieves performance comparable to the parameter-matched LorentzNet across Acc./AUC and Rej@30%, with differences within the reported uncertainty (i.e., overlapping standard deviations).

## I.2. Group Action Equivariance via Embedding Map

To process four-momenta within our Lie-algebraic framework, we require an embedding that translates the Lorentz action into an adjoint action on a matrix space.

**Definition I.1** (Lorentz-Compatible Embedding). Given $p \in \mathbb{R}^4$ and $\eta = \mathrm{diag}(-1, 1, 1, 1)$, define

$$\varphi(p) = \begin{bmatrix} 0_{4\times4} & p \\ p^\top \eta & 0 \end{bmatrix}. \tag{47}$$

**Theorem I.2** (Adjoint Equivariance). *For any $p \in \mathbb{R}^4$ and $\Lambda \in \mathrm{SO}^+(1,3)$, let $G = \mathrm{diag}(\Lambda, 1) \in \mathrm{GL}(5)$. Then*

$$\mathrm{Ad}_G(\varphi(p)) = G\varphi(p)G^{-1} = \varphi(\Lambda p). \tag{48}$$

*Proof.* We compute the left-hand side (LHS) of Eq. 48, which is the adjoint action:

$$\mathrm{Ad}_G(\varphi(p)) = \begin{bmatrix} \Lambda & 0 \\ 0 & 1 \end{bmatrix} \begin{bmatrix} 0 & p \\ p^\top \eta & 0 \end{bmatrix} \begin{bmatrix} \Lambda^{-1} & 0 \\ 0 & 1 \end{bmatrix} = \begin{bmatrix} 0 & \Lambda p \\ p^\top \eta \Lambda^{-1} & 0 \end{bmatrix}. \tag{49}$$

The right-hand side (RHS) is the lift of the transformed vector $\Lambda p$:

$$\varphi(\Lambda p) = \begin{bmatrix} 0 & \Lambda p \\ (\Lambda p)^\top \eta & 0 \end{bmatrix} = \begin{bmatrix} 0 & \Lambda p \\ p^\top \Lambda^\top \eta & 0 \end{bmatrix}. \tag{50}$$

For the LHS and RHS to be equal, we must show that $\eta \Lambda^{-1} = \Lambda^\top \eta$. We start from the defining property of $\mathrm{SO}(1,3)$:

$$\Lambda^\top \eta \Lambda = \eta. \tag{51}$$

Right-multiplying Eq. 51 by $\Lambda^{-1}$ yields the desired identity:

$$(\Lambda^\top \eta \Lambda)\Lambda^{-1} = \eta \Lambda^{-1} \implies \Lambda^\top \eta (\Lambda\Lambda^{-1}) = \eta \Lambda^{-1} \implies \Lambda^\top \eta = \eta \Lambda^{-1}. \tag{52}$$

Since the condition holds, the proof is complete. $\square$

*Remark* I.3 (Generalization to Orthogonal Groups). This embedding technique is not limited to the Lorentz group and can be readily generalized to any orthogonal group $\mathrm{O}(n)$ or special orthogonal group $\mathrm{SO}(n)$. For instance, in applications involving 3D point clouds where the symmetry is $\mathrm{SO}(3)$, a vector $p \in \mathbb{R}^3$ would be embedded into the Lie algebra $\mathfrak{gl}(4)$ as:

$$\varphi(p) = \begin{bmatrix} 0_{3 \times 3} & p \\ p^\top & 0 \end{bmatrix} \tag{53}$$

The proof of equivariance follows the same structure, using the property of orthogonal matrices, $R^\top R = I$ (which implies $R^{-1} = R^\top$), instead of the Minkowski metric identity. This highlights the broad applicability of our embedding strategy to any benchmark involving norm-preserving group transformations.

### I.3. Experimental Implementation

**Dataset.** The experiment uses the Top-Tagging dataset (Kasieczka et al., 2019), which contains 2 million simulated proton-proton collision events. The dataset was generated with Pythia, Delphos, and FastJet to model the ATLAS detector response. We use the standard 60%/20%/20% splits for training, validation, and testing. Each jet is represented as a set of constituent particles, each with four-momentum $p = (E, p_x, p_y, p_z)$.

**Model.** Our model leverages the established architecture of LorentzNet (Gong et al., 2022), utilizing its stack of Lorentz Group Equivariant Blocks (LGEBs) for message passing on the jet's particle cloud. While the original LorentzNet computes these features directly from the 4-momenta using the Minkowski inner product, our approach introduces a modified bilinear form-based feature extraction. We first embed each pair of 4-momenta, $p_i$ and $p_j$, from the Minkowski space $\mathbb{R}^{1,3}$ into the Lie algebra $\mathfrak{gl}(5)$ via the map $p \mapsto \varphi(p)$, as defined in Equation 47. The invariant features for the message passing are then derived from the bilinear form, $\widetilde{B}(\cdot, \cdot)$, on this Lie algebraic space. The edge message $m_{ij}$ is thus constructed as:

$$m_{ij} = \phi_e\Big(h_i, h_j, \psi\big(\widetilde{B}(\varphi(p_i), \varphi(p_i))\big), \psi\big(\widetilde{B}(\varphi(p_j), \varphi(p_j))\big), \psi\big(\widetilde{B}(\varphi(p_i), \varphi(p_j))\big)\Big) \tag{54}$$

where $h_i, h_j$ are scalar features, $\phi_e$ is an MLP, and $\psi$ is a stabilizing nonlinearity. As shown in the main results (Table 14), this approach achieves comparable background rejection to the parameter-matched LorentzNet baseline within the reported uncertainty. The architectural differences are summarized in Table 15.

*Table 15.* Architectural comparison for the Top-Tagging task.

| Component | LorentzNet (Original) | Param-matched Baseline | Ours (ReLN) |
|---|---|---|---|
| Number of LGEBs | 6 | 5 | 5 |
| Hidden feature dims | 72 | 48 | 48 |
| Edge feature computation | Minkowski inner prod. | Minkowski inner prod. | **Bilinear invariant form** |

**Training Setup.** For a fair comparison, our training procedure closely follows the protocol established in the LorentzNet (Gong et al., 2022). The model was trained for a total of 35 epochs on a NVIDIA RTX 4090 GPU. We used the AdamW optimizer with a weight decay of 0.01 and a batch size of 128, matching the total effective batch size from the reference work. The learning rate was managed by the paper's specific three-stage schedule: a 4-epoch linear warm-up to an initial rate of $1 \times 10^{-3}$, followed by a 28-epoch `CosineAnnealingWarmRestarts` schedule, and a final 3-epoch exponential decay. After each epoch, the model with the highest validation accuracy was saved for final evaluation on the test set.

## J. Drone Experiment Details

This appendix provides the technical details for the drone state estimation experiment, including the theoretical framework, dataset generation, model implementations, and formal proofs.

### J.1. Geometric Framework for Equivariant Covariance Processing

Our approach leverages the geometry of symmetric positive-definite matrices. A covariance matrix $C$ is symmetric positive-definite, residing on the manifold $\mathrm{SPD}(3)$. A non-degenerate covariance matrix $C \in \mathrm{SPD}(n)$ represents the anisotropic stretching of a general linear map, as seen via the polar decomposition $A = QP$ with $Q \in \mathrm{O}(n)$ and $P \in \mathrm{SPD}(n)$.

Equivalently, there is a homogeneous-space isomorphism: $\mathrm{SPD}(n) \cong \mathrm{GL}(n)/\mathrm{O}(n)$, which motivates processing covariances in a $\mathrm{GL}(n)$-aware architecture.

While $\mathrm{SPD}(3)$ is not a Lie group, the matrix logarithm provides a canonical map to the vector space of symmetric matrices $\mathrm{Sym}(3)$, which is a linear subspace of $\mathfrak{gl}(3)$.

$$\log : \mathrm{SPD}(3) \longrightarrow \mathrm{Sym}(3) \subset \mathfrak{gl}(3). \tag{55}$$

This allows us to embed a geometric object from a curved manifold into a flat, Lie-algebra-compatible space. The following theorem proves that the congruence transformation on $C \in \mathrm{SPD}(n)$ becomes an adjoint action on its image $\log C \in \mathrm{Sym}(n)$, thus preserving the equivariant structure required by our model.

**Theorem J.1** (Equivariance of the Logarithmic Map). *For any $C \in \mathrm{SPD}(n)$ and any rotation matrix $R \in \mathrm{SO}(n)$, the congruence transformation on $C$ corresponds to an adjoint action on its logarithm:*

$$\log(RCR^\top) = R(\log C)R^\top. \tag{56}$$

*Proof.* The proof follows from the spectral theorem for real symmetric matrices.

1. Let the eigendecomposition of $C$ be $C = V\Lambda V^\top$, where $V$ is an orthogonal matrix ($V^\top V = I$) of eigenvectors and $\Lambda$ is the diagonal matrix of corresponding positive eigenvalues.

2. By definition, the matrix logarithm of $C$ is given by applying the logarithm to its eigenvalues:

$$\log C := V(\log \Lambda)V^\top \tag{57}$$

where $\log \Lambda$ is the diagonal matrix of element-wise logarithms of the eigenvalues.

3. Consider the transformed matrix $C' = RCR^\top$. Substituting the decomposition of $C$ yields:

$$C' = R(V\Lambda V^\top)R^\top = (RV)\Lambda(V^\top R^\top) = (RV)\Lambda(RV)^\top \tag{58}$$

This is the eigendecomposition of $C'$, where the new orthogonal matrix of eigenvectors is $V' = RV$ and the eigenvalues $\Lambda$ are unchanged.

4. Applying the definition of the matrix logarithm to $C'$ gives:

$$\log(C') = V'(\log \Lambda)(V')^\top = (RV)(\log \Lambda)(RV)^\top \tag{59}$$

5. Rearranging the terms, we arrive at the desired identity:

$$\log(C') = R\left(V(\log \Lambda)V^\top\right)R^\top = R(\log C)R^\top \tag{60}$$

$\square$

This identity is critical, as it confirms that our adjoint-equivariant network can process either the raw covariance $C$ or its logarithm $\log C$ while perfectly preserving the $\mathrm{SO}(3)$ symmetry.

In the $\mathrm{SO}(3)$ regime used in our experiments, vectors (e.g., velocity $\mathbf{v}$) are represented in the Lie algebra $\mathfrak{so}(3)$ so that the adjoint action coincides with ordinary rotation, $\mathrm{Ad}_R(\mathbf{v}) = R\mathbf{v}$. Conjugation then implements the covariance congruence $C \mapsto RCR^\top$. Consequently, ReLNs realize $\mathrm{SO}(3)$-equivariance *by construction*, avoiding the need for the model to learn these symmetries from data.

## J.2. Dataset Generation.

We use the PyBullet engine to simulate 200 aggressive trajectories for a Crazyflie-like nano-quadrotor. To generate realistic measurements, the instantaneous velocity is corrupted by Gaussian noise, $\mathbf{v}_{\mathrm{noisy}} \sim \mathcal{N}(\mathbf{v}_{\mathrm{gt}}, C_v)$, where the covariance $C_v$ varies with flight aggressiveness. The dataset provides time series of noisy velocities, ground-truth covariances, and ground-truth trajectories for evaluation.

**Trajectory Generation.** The procedure begins with the procedural generation of a sequence of 20 to 40 random 3D waypoints within a flight volume of approximately $170\text{m} \times 170\text{m} \times 60\text{m}$. The waypoints are sampled from a uniform distribution to create diverse flight paths. To mimic the complex dynamics of aggressive flight, each trajectory is randomly generated using a path with random wiggles or a path featuring high-speed spiral maneuvers. These discrete waypoints are then interpolated using a Catmull-Rom spline to create a smooth, $C^1$ continuous target trajectory, which is densely sampled at an $80\,\text{Hz}$ control frequency. Each of the 200 sequences results in a unique trajectory lasting approximately 2-4 minutes, totaling over 13 hours of simulated flight time. A sample generated trajectory is shown in Figure 5.

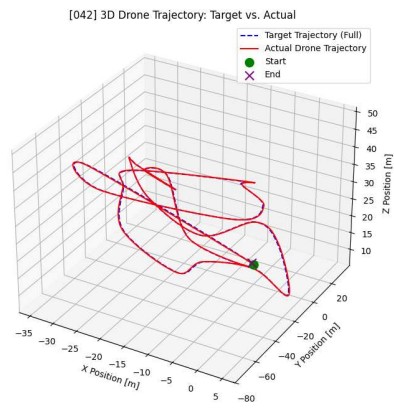 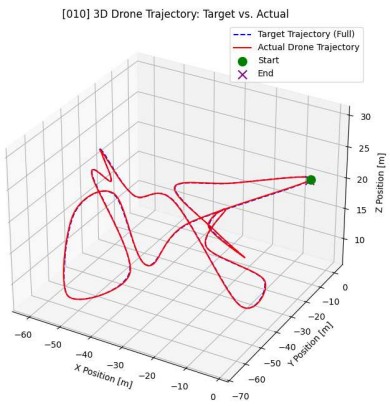

*(a)* A sample trajectory with spiral maneuvers.   *(b)* A sample trajectory with random wiggles.

*Figure 5.* Sample aggressive trajectories generated in the PyBullet simulator.

**State-Dependent Noise Model.** To simulate realistic sensor characteristics, the ground-truth velocity is corrupted by zero-mean Gaussian noise, $\mathbf{v}_{\text{noisy}} \sim \mathcal{N}(\mathbf{v}_{\text{gt}}, C_v)$. The covariance matrix $C_v$ is state-dependent, designed to scale with the drone's speed. The standard deviation $\sigma_v$ for each velocity axis is computed using a sigmoid function of the velocity magnitude $\|\mathbf{v}_{\text{gt}}\|$:

$$\sigma_v(\|\mathbf{v}_{\text{gt}}\|) = \sigma_{\min} + (\sigma_{\max} - \sigma_{\min}) \cdot \frac{1}{1 + \exp(-\lambda(\|\mathbf{v}_{\text{gt}}\| - v_{\text{mid}}))}, \tag{61}$$

where the variance on each axis is $\sigma_v^2$. We set the minimum and maximum standard deviations to $\sigma_{\min} = 0.2\,\text{m/s}$ and $\sigma_{\max} = 1.0\,\text{m/s}$, respectively. The steepness $\lambda$ is set to 0.8, and the midpoint velocity $v_{\text{mid}}$ is dynamically adjusted based on the estimated average speed of each trajectory to ensure a realistic noise profile.

### J.3. Baseline and Model Implementation Details

We compare ReLN against two baseline classes chosen to isolate the effect of geometric priors.

**Non-equivariant baselines.** We use a standard 1D ResNet architecture with temporal convolutional blocks that processes flattened input sequences. The **ResNet (velocity-only)** model receives only the 3D velocity vector. The **ResNet (velocity + covariance)** model receives the flattened $3 \times 3$ covariance matrix concatenated to the velocity vector.

**Eigendecomposition-based SO(3)-Equivariant Baseline.** This model adapts the 1D ResNet backbone for $SO(3)$ equivariance using VN layers. Since VNs cannot directly ingest matrices, we decompose each covariance matrix $C = V\Lambda V^\top$ and use a dual-stream design:

- an *equivariant* stream $\mathcal{F}_{\text{eq}} = \{\mathbf{v}, \mathbf{e}_1, \mathbf{e}_2, \mathbf{e}_3\}$ comprising the measured velocity $\mathbf{v}$ and the three orthonormal eigenvectors $\mathbf{e}_i$, which together capture all directional information. This stream is handled by the VNs backbone.

- an *invariant* stream $\mathcal{F}_{\text{inv}} = \{\lambda_1, \lambda_2, \lambda_3\}$ processes the corresponding eigenvalues $\{\lambda_1, \lambda_2, \lambda_3\}$, which encode orientation-independent scale information, using a standard MLP.

The two latent features from both streams are fused at the final output layer. Eigenvector ambiguities (sign or multiplicities) are resolved via a deterministic, rotation-equivariant canonicalization.

**Spherical Harmonic (SH)-based Equivariant Baselines (TFN and SE(3)-Transformer).** We evaluate two representative $SO(3)$-equivariant graph architectures built from irrep-valued features and steerable bases expanded in spherical harmonics (SH): Tensor Field Networks (TFN) (Thomas et al., 2018) and the SE(3)-Transformer (Fuchs et al., 2020). Both enforce equivariance by restricting linear maps to $SO(3)$ intertwiners, implemented through SH steerable kernels and tensor-product (Clebsch–Gordan) couplings.

*Transformation Rules and Irrep Decomposition.* Under a rotation $R \in SO(3)$, the velocity transforms as $\mathbf{v} \mapsto R\mathbf{v}$, while a covariance transforms by congruence $C \mapsto RCR^\top$. We represent the symmetric rank-2 covariance via the standard irrep decomposition $\mathrm{Sym}^2(\mathbb{R}^3) \cong L{=}0 \oplus L{=}2$, i.e., $C = \frac{\mathrm{tr}(C)}{3}I + C_0$ with $\mathrm{tr}(C)$ as a type-0 scalar and $C_0 := C - \frac{\mathrm{tr}(C)}{3}I$ as a type-2 traceless-symmetric tensor.

*Feature construction and temporal graph.* Each timestamp is encoded as an $SO(3)$ fiber with degrees $L \in \{0, 1, 2\}$: $\mathrm{tr}(C)$ for $L{=}0$, $\mathbf{v}$ for $L{=}1$, and $C_0$ for $L{=}2$. We form a temporal chain graph connecting $t$ to $t\pm 1$ and drive message passing using edge attributes derived from relative offsets. TFN aggregates messages via steerable kernel filtering (with learned radial profiles), whereas SE(3)-Transformer replaces fixed kernel aggregation with multi-head, content-dependent equivariant self-attention (Thomas et al., 2018; Fuchs et al., 2020). In both cases, uncertainty is stored in separate type-0 and type-2 blocks, and interactions between them are mediated by learned equivariant tensor-product couplings rather than by treating $C$ as a single matrix feature throughout the network.

*Relation to our formulation.* In contrast, ReLN embeds $(\mathbf{v}, C)$ into a unified matrix space $\mathfrak{gl}(3)$ and processes both under a single adjoint action, directly matching the congruence rule $C \mapsto RCR^\top$ in a shared Lie-algebraic backbone.

*Empirical observation.* In our experiments, TFN consistently outperformed the SE(3)-Transformer. We attribute this to the nature of drone state estimation as a path integration task. The SH-based convolutional kernels in TFN act as learned local integrators, which are better suited for capturing high-frequency temporal dynamics than the global dependency focus of attention mechanisms. While attention provides flexibility, it can introduce susceptibility to high-frequency noise in aggressive flight sequences, whereas the local inductive bias of SH-convolutions provides a more stable prior for trajectory reconstruction.

**Reductive Lie Neurons (ReLNs).** The ReLN model shares a similar backbone but incorporates the `ReLN-Bracket` layer. In contrast to the VN and graph-based baselines, ReLNs provide a unified framework for velocity and covariance processing within the Lie algebra $\mathfrak{gl}(3)$. Velocities $\mathbf{v} \in \mathbb{R}^3$ are lifted into $\mathfrak{so}(3) \subset \mathfrak{gl}(3)$ via $K = \mathbf{v}^\wedge$, while covariance $C$ (or $\log C$) is treated as a structured geometric input. Both transform under the same adjoint action: $K' = RKR^\top$ and $C' = RCR^\top$. By utilizing the matrix logarithm, $\log C \in \mathrm{Sym}(3) \subset \mathfrak{gl}(3)$, we embed the manifold-valued uncertainty into a linear subspace compatible with Lie-algebraic processing. The final velocity estimate is equivariantly extracted via the projection $\tilde{\mathbf{v}} = (\frac{1}{2}(A - A^\top))^\vee$ from the network's matrix output $A \in \mathbb{R}^{3 \times 3}$.

### J.4. Training and Evaluation Protocol

**Problem Formulation.** The network is trained to predict the drone's 3D position $\mathbf{p}_t \in \mathbb{R}^3$ at the end of a given time window, based on a sequence of noisy velocity measurements and their corresponding covariances within that window (e.g., a 1-second history). All models are trained by minimizing the Mean Squared Error (MSE) between the predicted position $\hat{\mathbf{p}}_t$ and the ground-truth position $\mathbf{p}_{t,\mathrm{gt}}$. The loss function is defined as $\mathcal{L} = \|\hat{\mathbf{p}}_t - \mathbf{p}_{t,\mathrm{gt}}\|_2^2$.

**Dataset and Optimization.** We partition the dataset using a standard 80:10:10 train/validation/test split. All models are trained on identical splits to ensure fair comparison. Models are optimized using the AdamW optimizer with a ReduceLROnPlateau learning rate scheduler based on validation loss.

**Evaluation Metrics.** We report the following pose-regression metrics over the test set:

- **Absolute Trajectory Error (ATE):** The root-mean-square error (RMSE) between the ground-truth positions $p_k$ and predicted 3D positions $\hat{p}_k$ over the entire trajectory, calculated as $\sqrt{\frac{1}{N} \sum_{k=1}^{N} \|\hat{p}_k - p_k\|^2}$, measured in meters.

- **ATE$_\%$:** The ATE normalized by the total trajectory length $L$, expressed as a percentage ($100 \times \text{ATE}/L$). This metric provides a scale-invariant measure of error, which is crucial for fairly comparing performance across our aggressive flight trajectories of varying lengths.

- **Relative Translation Error (RTE):** This metric evaluates local consistency by measuring the relative translational error over fixed-length sub-trajectories. For each window of duration $\Delta t = 2.0\,\text{s}$ ($n = 400$ samples), we compute the RMSE of the positional difference between estimated and ground-truth relative displacements: $\|(\hat{p}_{k+n} - \hat{p}_k) - (p_{k+n} - p_k)\|$. We report the mean error by sliding this window across the trajectory with a stride of $0.5\,\text{s}$ (100 samples) to ensure a robust assessment of short-term stability.

To explicitly validate equivariance, we also evaluate all models on the test set after applying a set of random $SO(3)$ rotations to the entire input sequence.

### J.5. Eigenvector Canonicalization for the VN Baseline

To resolve ambiguities in the eigendecomposition $C = V\Lambda V^\top$ for the VN baseline while preserving $SO(3)$ geometry, we canonicalize the eigenvector matrix $V = [\mathbf{e}_1, \mathbf{e}_2, \mathbf{e}_3]$ using the following sequence. To avoid producing reflection matrices (where $\det V = -1$), sign disambiguation must precede the enforcement of a right-handed frame.

1. **Sign Disambiguation:** For each eigenvector $\mathbf{e}_i$ associated with a distinct eigenvalue, we enforce a consistent orientation relative to the velocity vector $\mathbf{v}$ by ensuring $\mathbf{v}^\top \mathbf{e}_i \geq 0$. If $\mathbf{v}^\top \mathbf{e}_i < 0$, we set $\mathbf{e}_i \leftarrow -\mathbf{e}_i$.

2. **Right-handed Frame:** After the individual signs are fixed, we ensure the matrix $V$ represents a proper rotation. If $\det V < 0$, we flip the sign of the third eigenvector, $\mathbf{e}_3 \leftarrow -\mathbf{e}_3$, to ensure $\det V = +1$. This step is performed last to ensure that any sign changes in the previous step do not inadvertently result in a reflection frame.

3. **Multiplicity Handling:** In cases of repeated eigenvalues, we project the velocity vector $\mathbf{v}$ onto the corresponding eigenspace to uniquely define the first basis vector of that subspace, then complete the orthonormal basis via the Gram-Schmidt process, followed by the determinant check in Step 2.

By applying the determinant constraint after sign alignment, we guarantee that the baseline operates on valid rotation matrices, ensuring a fair and geometrically consistent comparison with our $SO(3)$-equivariant model.

### J.6. Full Experimental Results and Ablation Study for Drone State Estimation

Table 16 provides the exhaustive results for all models across three input modalities: $v, (v, C), (v, \log C)$. This comprehensive comparison highlights that while the $\log$-covariance interface generally improves performance for most equivariant models, our ReLN architecture remains the superior backbone due to its ability to jointly model the full reductive structure.

**Qualitative Analysis of Trajectory Fidelity.** Figure 6 validates the qualitative robustness of ReLNs under high-dynamic flight conditions. While standard MLP-based ResNets and eigendecomposition-based Vector Neurons suffer from rapid error accumulation and structural divergence, ReLN maintains high fidelity to the ground truth even in velocity-only configurations.

The integration of covariance, $(v, C)$, and specifically the $\log$-**covariance** provides effective regularization via uncertainty-aware weighting. Notably, while steerable baselines such as TFN and SE(3)-Transformers show improved stability with $\log$-covariance, they still exhibit observable deviations during sharp maneuvers where geometric consistency is critical. This confirms our finding that the reductive Lie-algebraic backbone offers a more stable and geometrically principled space for fusing heterogeneous sensor signals and their associated uncertainties.

# K. Proof of $SO(3)$-Equivariance for ReLN Velocity Extract with Covariance Inputs

This section provides a formal proof for the $SO(3)$-equivariance of our Reductive Lie Neuron (ReLN) architecture when processing a velocity vector and a covariance matrix. We first establish the foundations for processing covariance matrices within a Lie-algebraic framework and then present the main proof.

*Table 16.* Complete performance report on the drone dataset including all input variants and ablations. We report ATE (meters), ATE$_\%$ (scaled), and RTE (meters). For the headline configurations of each model family, we additionally report the standard deviation across 3 seeds on ATE and RTE; entries without $\pm$ are single-seed runs. **SO(3)** denotes evaluation under random test-time rotations.

| Model | Input | ID (In-Distribution) | | | SO(3) (Rotated) | | |
|---|---|---|---|---|---|---|---|
| | | ATE $\downarrow$ | ATE$_\%$ $\downarrow$ | RTE $\downarrow$ | ATE $\downarrow$ | ATE$_\%$ $\downarrow$ | RTE $\downarrow$ |
| ResNet | $v$ | 208.07 | 95.06 | 107.60 | 217.02 | 100.39 | 111.29 |
| ResNet | $(v, C)$ | $205.11 \pm 2.4$ | 94.94 | $106.07 \pm 1.5$ | $213.26 \pm 2.4$ | 98.90 | $109.37 \pm 1.5$ |
| Vector Neurons | $v$ | $17.36 \pm 0.4$ | 7.52 | $13.51 \pm 0.1$ | $17.36 \pm 0.4$ | 7.52 | $13.51 \pm 0.1$ |
| Vector Neurons | $(v, C)$ | 191.78 | 88.66 | 98.39 | 190.22 | 88.47 | 98.26 |
| Tensor Field Network | $v$ | 24.59 | 10.95 | 18.23 | 24.59 | 10.95 | 18.23 |
| Tensor Field Network | $(v, C)$ | 17.56 | 7.60 | 14.40 | 17.56 | 7.60 | 14.40 |
| Tensor Field Network | $(v, \log C)$ | $16.83 \pm 0.6$ | 7.56 | $13.34 \pm 0.7$ | $16.83 \pm 0.6$ | 7.56 | $13.34 \pm 0.7$ |
| SE(3)-Transformer | $v$ | 22.22 | 9.85 | 17.63 | 22.22 | 9.85 | 17.63 |
| SE(3)-Transformer | $(v, C)$ | 21.67 | 9.37 | 16.77 | 21.67 | 9.37 | 16.77 |
| SE(3)-Transformer | $(v, \log C)$ | 20.12 | 8.84 | 15.36 | 20.12 | 8.84 | 15.36 |
| *Ablations (Ours)* | | | | | | | |
| Lie Neurons[†] | $(v, C)$ | 16.86 | 7.43 | 13.65 | 16.86 | 7.43 | 13.65 |
| Lie Neurons[†] | $(v, \log C)$ | $15.65 \pm 0.1$ | 6.76 | $12.04 \pm 0.1$ | $15.65 \pm 0.1$ | 6.76 | $12.04 \pm 0.1$ |
| ReLN (no semisimple) | $(v, \log C)$ | 16.27 | 7.00 | 12.65 | 16.27 | 7.00 | 12.65 |
| *Our Equivariant Model* | | | | | | | |
| ReLN | $v$ | 16.85 | 7.31 | 12.70 | 16.85 | 7.31 | 12.70 |
| ReLN | $(v, C)$ | 16.49 | 7.21 | 13.02 | 16.49 | 7.21 | 13.02 |
| ReLN | $(v, \log C)$ | $\mathbf{13.92 \pm 0.6}$ | $\mathbf{5.99}$ | $\mathbf{11.04 \pm 0.5}$ | $\mathbf{13.92 \pm 0.6}$ | $\mathbf{5.99}$ | $\mathbf{11.04 \pm 0.5}$ |

[†] The Lie Neurons architecture (Lin et al., 2024a) is mathematically equivalent to the "no-center" variant of ReLN, i.e., ReLN with the bilinear form restricted to the semisimple ideal.

## K.1. SO(3)-Equivariant Vector Extraction via Skew-Symmetric Projection

Our network, $\Phi$, is designed to be adjoint-equivariant. It maps geometric inputs—such as an embedded velocity $K \in \mathfrak{so}(3)$ and a covariance matrix $S \in \text{SPD}(3)$—to a matrix feature $A \in \mathbb{R}^{3 \times 3}$. The inputs transform under the adjoint action of any rotation $R \in \text{SO}(3)$:

$$K' = \text{Ad}_R(K) = RKR^\top, \quad S' = \text{Ad}_R(S) = RSR^\top. \tag{62}$$

By construction, the network's output feature $A$ transforms according to the same law:

$$\Phi(K', S') = \text{Ad}_R\big(\Phi(K, S)\big) = R\,\Phi(K, S)\,R^\top. \tag{63}$$

To obtain the final 3D velocity vector, we project the output matrix $A$ onto its skew-symmetric component and apply the vee operator. The following proposition formalizes the equivariance of this extraction mechanism.

**Proposition K.1** (Equivariance of Skew-Symmetric Extraction). *Let a network $\Phi$ and its inputs transform according to Eqs. 62 and 63. If a vector $\tilde{\mathbf{v}} \in \mathbb{R}^3$ is extracted from the output matrix $A = \Phi(K, S)$ via the projection*

$$A_{\text{skew}} = \tfrac{1}{2}(A - A^\top), \qquad \tilde{\mathbf{v}} = (A_{\text{skew}})^\vee, \tag{64}$$

*then the vector $\tilde{\mathbf{v}}'$ extracted from the transformed output $A' = \Phi(K', S')$ transforms covariantly as $\tilde{\mathbf{v}}' = R\tilde{\mathbf{v}}$.*

*Proof.* By the adjoint-equivariance property in Eq. 63, the network satisfies $\Phi(RKR^\top, RSR^\top) = R\,\Phi(K, S)\,R^\top = RAR^\top$. Let $A' = RAR^\top$. The skew-symmetric component of the transformed output $A'$ is:

$$\begin{aligned}
A'_{\text{skew}} &= \tfrac{1}{2}(A' - A'^\top) \\
&= \tfrac{1}{2}\big(RAR^\top - (RAR^\top)^\top\big) \\
&= \tfrac{1}{2}\big(RAR^\top - RA^\top R^\top\big) \\
&= R\left(\tfrac{1}{2}(A - A^\top)\right)R^\top \\
&= RA_{\text{skew}}R^\top = \text{Ad}_R(A_{\text{skew}}).
\end{aligned} \tag{65}$$

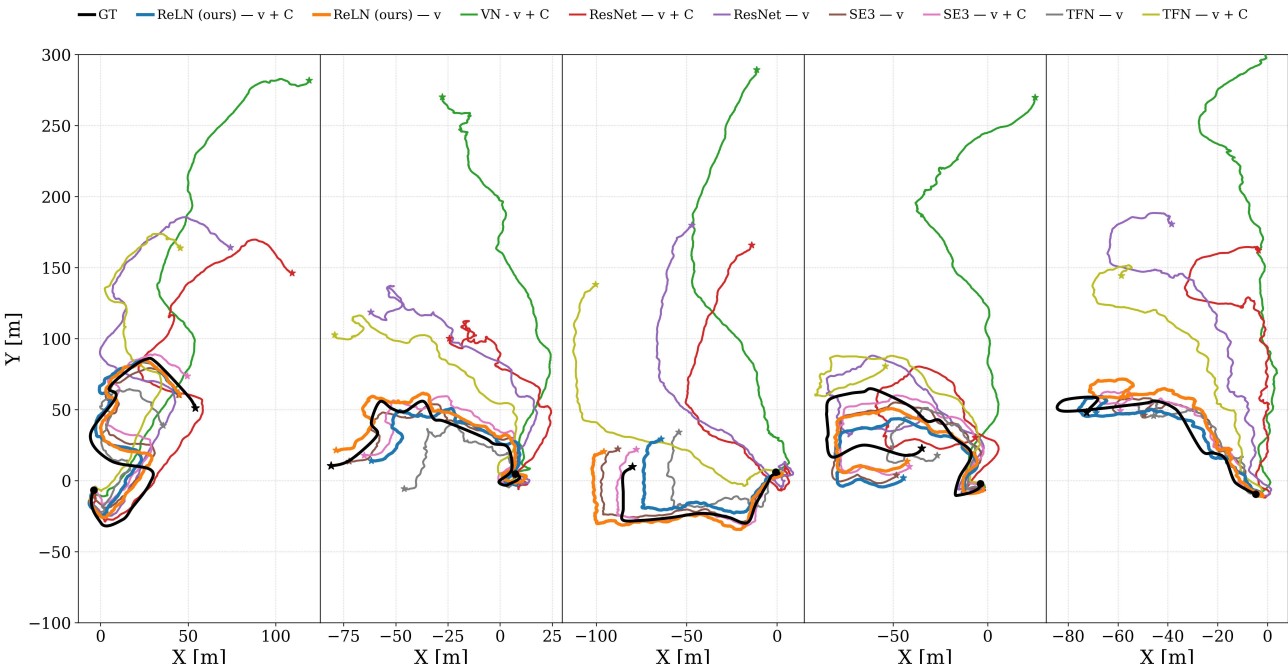

*Figure 6.* Qualitative comparison of reconstructed trajectories ReLN variants maintain superior tracking fidelity and stability compared to non-equivariant and standard equivariant baselines.

The vee map, $(\cdot)^\vee : \mathfrak{so}(3) \to \mathbb{R}^3$, is itself an equivariant map satisfying $(\mathrm{Ad}_R(X))^\vee = R\,(X^\vee)$ for any $X \in \mathfrak{so}(3)$. Applying this property yields the desired result:

$$\tilde{\mathbf{v}}' = (A'_{\mathrm{skew}})^\vee = (\mathrm{Ad}_R(A_{\mathrm{skew}}))^\vee = R\,(A_{\mathrm{skew}})^\vee = R\tilde{\mathbf{v}}. \tag{66}$$

$\square$

*Remark* K.2. The proof relies on three properties: (i) both inputs transform under the adjoint action $X \mapsto RXR^\top$; (ii) the network $\Phi$ is equivariant to this action; and (iii) the output is projected onto $\mathfrak{so}(3)$ before the vee operator is applied. As established previously, these conditions hold whether the network ingests the raw covariance $S$ or its logarithm $\log S$.

## L. Details on 3D Gaussian Splatting Experiments

In this section, we detail the experimental protocols and architectural modifications for the 3D Gaussian Splatting (3DGS) experiments. Figure 7 provides a comprehensive overview of the ReLN-integrated Gaussian-MAE architecture, highlighting the equivariant data flow from geometric lifting to projection. We provide a rigorous theoretical justification for our architectural design to achieve **end-to-end equivariance** by jointly learning geometric and photometric features within a unified Lie-algebraic framework.

### L.1. Datasets and Protocol

We adopt a standard two-stage protocol common in self-supervised learning for 3D representation (Ma et al., 2025), separating the pre-training domain from the downstream evaluation task to assess generalization.

**Pre-training: ShapeSplat (ShapeNet-derived).** We utilize **ShapeSplat** (Ma et al., 2025), a large-scale collection of object-level 3D Gaussian primitives derived from ShapeNet. Each object is represented by a set of anisotropic Gaussians $\Theta_i = \{\mu_i, \Sigma_i, \alpha_i, c_i\}_{i=1}^N$. Following the baseline protocol, we sample $N = 1,024$ Gaussians via furthest point sampling (FPS) and apply a $60\%$ masking ratio for the masked autoencoding objective.

**Downstream: ModelNet10 Classification.** To evaluate rotational robustness, we fine-tune the pre-trained encoder on **ModelNet10**. We report accuracy on: (i) **Aligned**: The canonical test set. (ii) **Rotated**: The test set under random rotations

$R \sim SO(3)$, testing whether the learned representations remain stable under arbitrary orientations.

## L.2. Holistic Equivariance via Joint Feature Learning

A critical limitation of the baseline architecture (Ma et al., 2025) is the naive concatenation of heterogeneous Gaussian attributes. A 3D Gaussian combines *equivariant* quantities ($\mu, \Sigma$) with *invariant* quantities ($c, \alpha$). Our goal in ReLN is to construct a **holistic equivariant architecture** where these distinct attribute types are learned jointly without violating the symmetry of the feature space.

**1. The Equivariant Component (Active Geometry).** The mean position $\mu \in \mathbb{R}^3$ and covariance $\Sigma \in \mathbb{R}^{3\times3}$ dictate the spatial structure. Under a global rotation $R \in SO(3)$, these transform via $\mu \mapsto R\mu$ and $\Sigma \mapsto R\Sigma R^\top$. Recognizing that the covariance update corresponds to the adjoint action $\mathrm{Ad}_R(\Sigma) = R\Sigma R^{-1}$, we embed $\Sigma$ into the symmetric subspace of $\mathfrak{gl}(3)$. This ensures that the primary geometric features are processed within a consistent $\mathrm{Ad}$-equivariant vector space throughout the network.

**2. The Invariant Component (Photometry and Geometric Norms).** To support robust representation learning, we augment the state space with a set of $O(3)$-**invariant scalars (Type-0 features)**. This set includes:

- **Intrinsic Attributes.** Opacity $\alpha$ and color $c$ are treated as rotation-invariant channels in our setup. Following the baseline, we retain only the *zeroth-order* spherical-harmonic term ($l = 0$) for color (three coefficients for the RGB channels). Concretely, an SH color field can be written as $f(\hat{\mathbf{d}}) = \sum_{l,m} c_{lm} Y_l^m(\hat{\mathbf{d}})$, and for $l = 0$ this reduces to $f(\hat{\mathbf{d}}) = c_{00} Y_0^0(\hat{\mathbf{d}})$. Since $Y_0^0$ is constant on the sphere, the $l = 0$ coefficient is unchanged by rotations (equivalently, the Wigner-$D$ matrix satisfies $D^0(R) = 1$). Therefore, the RGB $l = 0$ coefficients are best viewed as *three independent scalar channels* (i.e., multiplicity-3 copies of the $L = 0$ irrep), and are invariant under object rotations: $c \mapsto c$ and $\alpha \mapsto \alpha$.

- **Derived Geometric Invariants:** Crucially, we also explicitly compute rotation-invariant geometric descriptors, such as the norms of position vectors ($\|\mu\|$) and rotation/scale magnitudes derived from $\Sigma$.

In the ReLN framework, treating these as Type-0 features serves to *facilitate its feature integration*. These scalars participate in the learning process by modulating the equivariant features without violating the underlying group structure and equivariance.

**3. Joint Learning in ReLN Attention.** The explicit stratification of these streams is a structural prerequisite for **joint equivariant learning**. By assigning each attribute to its correct algebraic type (Type-1/Type-2 for geometry, Type-0 for scalars), the ReLN attention mechanism can validly integrate information across all modalities. The invariant scalars $(c, \alpha, \|\mu\|)$ contribute to the computation of attention scores (invariant inner products), thereby modulating how the equivariant geometric features ($\mu, \Sigma$) are aggregated. This design ensures that the entire network optimizes a single, cohesive objective function while guaranteeing that the final representation remains equivariant end-to-end.

## L.3. Architectural Modifications

To enforce this holistic equivariance, we replaced non-robust components of the baseline architecture with ReLN equivalents that respect the defined type system.

**ReLNEncoder and Geometric Aggregation.** Our **ReLNEncoder** begins with a *Geometric Lifting* stage that maps raw Gaussian parameters $(C, S, R)$ into $\mathfrak{gl}(3)$. The baseline uses a 'SoftEncoder' (MLP on raw coordinates) which breaks equivariance because standard linear weights $W$ do not rotate with the input frame. We replaced this with **ReLNEncoder**, which employs $\mathrm{Ad}$-equivariant linear layers (ReLNLinear) to process local neighborhoods. ReLNEncoder incorporates a *Geometric Aggregator*, which utilizes an equivariant norm-based max pooling mechanism to compress grouped splats into structured $C' \times 9$ tokens. This ensures that the fundamental geometric tokens are processed within a mathematically consistent vector space.

**Central Positional Embedding.** To provide spatial context without violating symmetry, we process group centroids ($\mu$) through a dedicated *Central Positional Embedding*. This module lifts 3D spatial coordinates into a structured Lie-algebraic

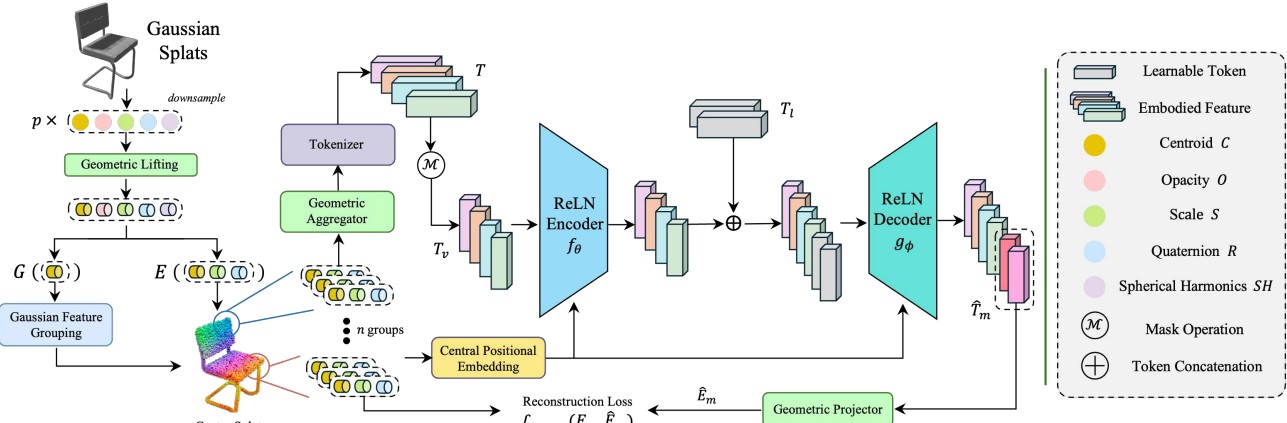

*Figure 7.* **The ReLN-integrated Gaussian-MAE Framework.** Our architecture pipelines raw 3D Gaussian splats through a **Geometric Lifting** stage to map them into the $\mathfrak{gl}(3)$ Lie algebra. The model explicitly bifurcates the data flow: (1) **Active Geometry** (position $\mu$, covariance $\Sigma$) is processed via the **ReLN Encoder** and **Decoder** to maintain equivariance, while (2) **Invariant Attributes** (color $c$, opacity $\alpha$) are integrated as Type-0 features. A dedicated **Central Positional Embedding** ensures stable spatial context. Finally, the **Geometric Projector** utilizes the Vee map and the modified Ad-invariant bilinear form $\widetilde{B}$ to reconstruct the physical Gaussian parameters while preserving holistic symmetry.

representation, which is then injected into both the encoder and decoder stages (as illustrated in Figure 7). This injection strategy ensures that the transformer blocks distinguish relative spatial relations while maintaining rotational equivariance across the entire processing pipeline.

**Invariant Global Pooling (vs. CLS Token).**  For downstream classification tasks, we eschew the conventional learnable `[CLS]` token, recognizing that a fixed-reference-frame parameter **inherently imposes** a canonical orientation bias that violates rotational symmetry. Instead, we employ a permutation-invariant *Global Max Pooling* over the final sequence of equivariant tokens. This ensures that the aggregated global representation remains stable and rotates consistently with the input geometry, thereby satisfying holistic rotational symmetry for robust 3D representation learning.

### L.4. Training and Evaluation Protocol

**Pre-training Setup.** We pre-trained both the baseline and ReLN models for 300 epochs on the ShapeSplat (ShapeNet) dataset. We utilized the AdamW optimizer with an initial learning rate of $1 \times 10^{-3}$, a weight decay of $0.05$, and a cosine annealing scheduler with a 10-epoch warm-up. For the ReLN model, we explicitly disabled the `soft_knn` option to prevent the leakage of non-equivariant features from the legacy encoder implementation.

**Fine-tuning Setup.** For the downstream ModelNet10 classification task, we fine-tuned the pre-trained encoder for an additional 300 epochs. In this phase, we adjusted the learning rate to $5 \times 10^{-4}$ while maintaining the same weight decay $(0.05)$ and cosine scheduling strategy.

**Evaluation Metrics.** To rigorously evaluate geometric stability, we construct two test benchmarks:

- **Standard (Aligned):** The original ModelNet10 test set where objects are canonically aligned.

- **Rotated (SO(3)):** We apply a random rotation $R \in \mathrm{SO}(3)$ to every object. For a Gaussian splat, we transform parameters as $\mu' = R\mu$, $\Sigma' = R\Sigma R^\top$.

### L.5. Additional Convergence Results

Figure 8 presents the detailed training dynamics. The ReLN model consistently achieves lower loss values across all geometric attributes—Rotation, Scale, and Spherical Harmonics (SH)—demonstrating that enforcing equivariance stabilizes the learning of intrinsic 3D shape descriptors.

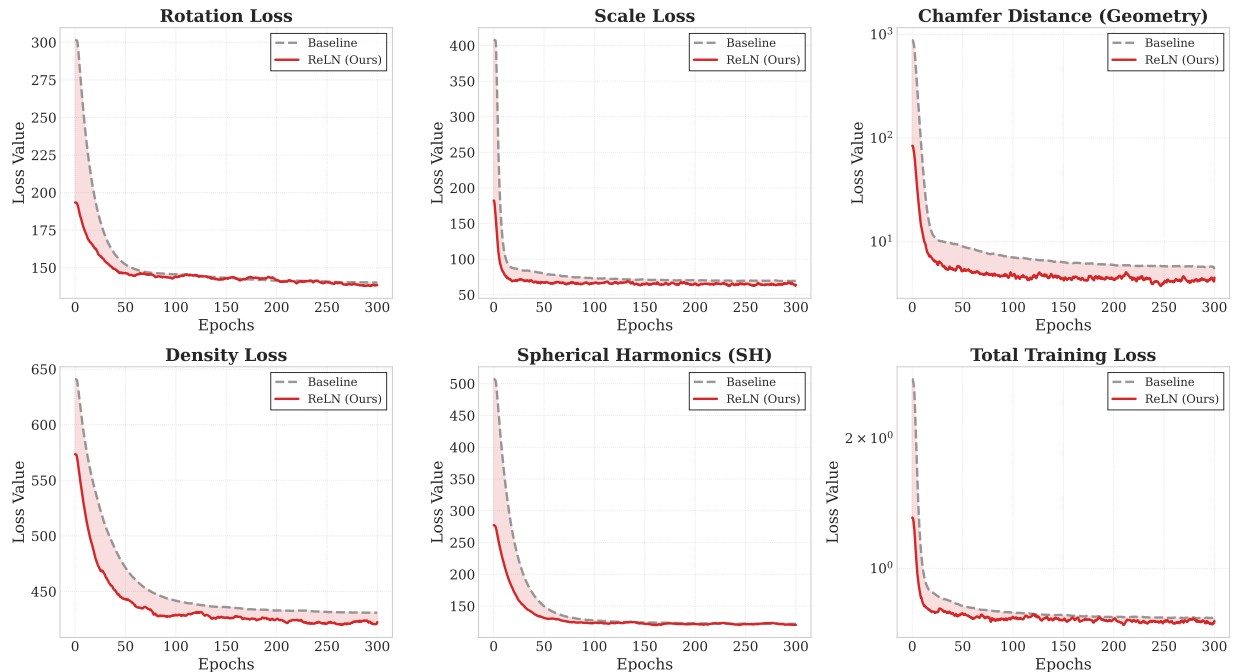

*Figure 8.* **Pre-training Dynamics on 3D Gaussian Splats.** Comparison of training convergence between the Baseline (Ma et al., 2025) and ReLN (Ours) across 300 epochs on **ShapeNet**. The red shaded regions highlight the gap where ReLN achieves lower reconstruction error than the baseline. ReLN consistently outperforms the baseline across **all** tracked metrics (including Rotation, Scale, Density, and Chamfer Distance), demonstrating that equivariance improves every aspect of 3D Gaussian reconstruction.

## M. Experimental Details for the Double-Pendulum Benchmark

To ensure a rigorous and fair comparison, we adhere to the experimental protocol and system parameters of the EMLP benchmark (Finzi et al., 2021).

**System dynamics and dataset generation.** We learn Hamiltonian dynamics of a 3D double spring pendulum exhibiting $O(2)$ symmetry about the $z$-axis. All physical constants are set to unity ($m_i = k_i = l_i = 1.0$) with gravity $g = [0, 0, 1]$. Ground-truth trajectories are generated by Hamiltonian integration over a horizon of $T = 30$s. We construct 1500 trajectory chunks of duration 1s (sampled at five 0.2s intervals) and split them into $500/500/500$ for training/validation/testing.

**Models and training.** All compared Hamiltonian models parameterize a scalar Hamiltonian $H_\theta(z)$ and use the standard HNN update $\dot{z} = J\nabla_z H_\theta(z)$, where $z$ stacks generalized coordinates and momenta and $J$ is the canonical symplectic matrix. Both EMLP-HNN and ReLN-HNN use three hidden layers with channel width $c = 128$, and are trained for 2000 epochs with Adam (lr=$3 \times 10^{-3}$, batch size=500). Rollout performance is evaluated by the geometric mean of relative errors over the full 30s horizon, averaged over three independent trials (random seeds).

**EMLP-HNN baseline (equivariant layer construction).** Let $G = O(2)$ be the symmetry group acting on the $(x, y)$ plane (with $z$ unchanged), e.g., $R(\phi) = \text{diag}(R_{2\times2}(\phi), 1)$ (and similarly for reflections), and let $\rho(\cdot)$ denote the induced representation on the state space. Equivariance of a linear map $W : V_{\text{in}} \to V_{\text{out}}$ requires

$$\rho_{\text{out}}(g)\, W = W\, \rho_{\text{in}}(g), \qquad \forall g \in G. \tag{67}$$

Following (Finzi et al., 2021), EMLP enforces this by constraining the vectorized weights $v = \text{vec}(W)$ under the product representation $\rho = \rho_{\text{out}} \otimes \rho_{\text{in}}^*$:

$$\rho(g)\, v = v, \qquad \forall g \in G. \tag{68}$$

To avoid infinitely many constraints, the condition is reduced to a finite generator set consisting of Lie algebra generators $\{A_i\}$ and discrete generators $\{h_j\}$:

$$d\rho(A_i)\, v = 0 \quad \text{and} \quad (\rho(h_j) - I)\, v = 0, \tag{69}$$

which are assembled into a linear system $Cv = 0$. An equivariant basis $Q$ is obtained by computing a nullspace of $C$ (e.g., via SVD/Krylov methods as in the reference implementation), and the layer is parameterized as $v = Q\beta$ during training (no further constraint solving during SGD). This basis computation is group- and representation-specific (e.g., $O(2)$ vs. $SO(2)$).

**EMLP nonlinearity.** We use the standard EMLP gated nonlinearity (Finzi et al., 2021), where scalar channels produce gates that multiplicatively modulate non-scalar feature blocks, preserving equivariance while enabling expressive nonlinear mixing.

**ReLN-HNN (ours).** In contrast, ReLN-HNN bypasses numerical constraint solving and constructs equivariant/invariant primitives in closed form within a reductive Lie-algebraic space (here $\mathfrak{gl}(3)$), using bilinear/Killing-form-based operations to obtain symmetry-consistent features without group-specific basis recomputation. The same architecture is used across different symmetry settings (e.g., $O(2)$, $SO(2)$, $D_6$) without re-solving linear constraints.

**Computational efficiency measurement.** Per-step FLOPs in Table 7 are measured using JAX HLO cost analysis, which counts device-executed floating-point operations including internal tensor contractions and projections.

