# OpenReview forum: "Equivariant Neural Networks for General Linear Symmetries on Lie Algebras"
_ICML.cc/2026/Conference — ICML 2026 regular_

### Official Review · Reviewer_jS9p · 2026-02-16

**Soundness:** 3
**Presentation:** 3
**Significance:** 3
**Originality:** 2
**Overall Recommendation:** 4
**Confidence:** 4

**Summary:**

The authors introduce ReLNs, an architecture designed for $\mathrm{GL}(n)$ equivariance. Features with different transformation properties are embedded in the Lie algebra $\mathfrak{gl}(n)$, and equivariance is obtained by constructing layers which are manifestly equivariant under the adjoint action of $\mathrm{GL}(n)$ on $\mathfrak{gl}(n)$.

The main theoretical contribution is the construction of an Ad-invariant bilinear form on reductive, but not necessarily semisimple, Lie algebras. The form is symmetric and non-degenerate, which allows the authors to generalise certain nonlinearities and pooling/invariant layers to the case where the Killing form of the symmetry group is degenerate.

Following the construction of the ReLNs, the authors apply the framework to multiple problems with heterogeneous features and compare their model with non-equivariant, augmented, and equivariant benchmarks. The results verifies the theoretical properties of the model and indicate that it's competitive with the closest related Lie-algebra based benchmark (Lin et al. 2024).

**Compliance With Llm Reviewing Policy:**

Affirmed.

**Final Justification:**

In their rebuttal, the authors addressed my concern regarding the limited theoretical analysis, and clarified the claims and how the experiments support those claims. When incorporated in the paper, this will improve the soundness of the paper and make it more suitable for publication. The limited theoretical novelty remains an issue, however, which is the reason my recommendation is a weak accept.

**Key Questions For Authors:**

1. Can you elaborate on how degeneracy of the Killing form makes the Lie Neuron ill-conditioned and what "effectively linear" means in this context? Also, I don't quite understand where the non-degeneracy of the $\tilde{B}$-form enters in the theoretical construction. Can you please clarify where it does, or if it doesn't why not?

2. Can you clarify the statement in 5.2 that the decomposition into irreps ignores some aspect of the geometry connecting fields transforming in different reps.? Can you show theoretically that the approach of embedding into the Lie algebra and encoding equivariance as Ad-equivariance respects this structure?

3. Can you please clarify how the experimental results demonstrate "strong robustness gains" as claimed in the main contribution statement?

**Limitations:**

Yes.

**Strengths And Weaknesses:**

Summary:
Overall, I found the theoretical approach of the paper appealing. Extending previous Lie-algebraic architectures to GL(n) (or generally reductive Lie algebras) by constructing a non-degenerate extension of the Killing form is natural idea motivated by applications. However, there is no strong theoretical argument for why non-degeneracy should improve performance, and the experimental results seem to indicate marginal (if any) improvement even in the cases motivating the new architecture. The quality of the presentation also suffers in places, and the representation of experimental results is not entirely accurate.

Soundness:
The theoretical claims regarding the properties of the modified Killing form $\tilde{B}$ seem to be accurate and the proofs correct. However, the majority of the paper is dedicated to demonstrating that the use of $\tilde{B}$ to construct non-linearities and contractions translates into superior performance compared to the Lie Neuron (Lin et al. 2024). I find the experimental results unconvincing. Improvements compared to Lie Neuron are marginal at best. In particular, in the cases where error estimates are available the authors incorrectly claim that their model performs better, when the figures stated (if I read them correctly) explicitly precludes this conclusion (this applies to results presented in Tables 1 and 5). In Table 13, the results do indicate increased performance as claimed, but the lack of error estimates makes it hard to judge the quantitative gain.

The selection of results to include in the main paper is also somewhat strange. In Table 3, the best performing Lie Neuron (v, log C) from Table 13 is excluded. This is probably an oversight, but the selection of trajectories in Figure 4 is more problematic. From Figure 6 it seems that only the models known to perform poorly (ResNets and VN v+C) are included, whereas the models that perform competitively in Table 3 are excluded. Indeed, it is very hard to conclude systematic differences between the trajectories of the ReLN and the more competitive models by inspection in Figure 6. Furthermore, trajectories for the Lie Neuron main competitor are not presented anywhere. There might be a good reason for this selection of results, but it must definitely be justified.

Presentation:
The paper is reasonably well structured and clearly written overall. The author clearly describes the relation of the work to many relevant previous results. The quality of the presentation in section 4 is not sufficient; the description of the different layers in the framework is hard to parse, and the relation (if any) to the Lie Neuron (Lin et al. 2024) should be clearly stated. I would suggest expanding the presentation to help the reader understand the embedding into $\mathfrak{g}$, the way the GL(n) equivariance is accomplished through Ad-equivariance in this embedding, and the distinction between geometric left action and linear right action. A discussion summarising the experimental results and explaining how to interpret them would also be helpful, in particular relative to the claims made in the introduction. The symmetry groups relevant for the different problems, and whether they require the general GL(n) framework, could be more clearly discussed to further help the reader interpret the rationale behind the experimental design and interpretation of results.

Significance:
The main problem addressed is important, and the ambition to provide a more solid foundation for equivariant models in the GL(n) case is highly relevant. The authors argue for this primarily based on several important applications. A solid theoretical framework for GL(n) symmetries would also be important as a unifying description of all matrix Lie groups. Unfortunately, the actual contribution is focused on experimental investigation and falls somewhat short of this ambition.

Originality:
The theoretical developments does contribute to a better understanding of the algebraic foundations of Lie-algebraic models in the case when the algebra is not semisimple. The conceptual contribution compared to previous models, however, seems to be limited. If I understand correctly, all layers in the ReLN are inherited from the Lie Neuron, which is similarly based on the existence of a suitable non-degenerate form (the Killing form for the semisimple case).

Major issues/suggestions/questions:
* One thing that would greatly improve the paper would be a theoretical analysis comparing the LN and ReLN models and contrasting them with the steerable baselines. In particular, of course, for the case of degenerate Killing form. Statements to this effect are currently not specific, and there is no comprehensive theoretical argument for why the ReLN should perform better. To me, such a discussion would be much more interesting and convincing than the current extensive experimental investigation.
* The discussion of non-compactness in the introduction is somewhat confusing. The main obstruction for GL(n) is that the Killing form is degenerate. E.g. SL(2,R) is also non-compact but is simple and therefore has a non-degenerate (but indefinite) Killing form. Furthermore, the taxonomy in Figure 2 is not entirely clear; E(n) is neither reductive nor compact and is affine rather than linear. The authors should carefully clarify the connection between the different properties of Lie algebras discussed in the paper.
* Is the construction of the modified Killing form known in Lie algebra theory? If so, references to appropriate sources should be added.
* Why is the paper on equivariant CNNs on homogeneous spaces by Cohen et al. not discussed in the related works? If I recall correctly, they restrict to compact groups, but I don't think there is a restriction to semisimple groups.

Minor issues/suggestions/questions:
* Incomplete sentence L43R
* Formatting in Def 4.1: parentheses and equation number.
* Table captions should be self-contained and provide enough information to interpret the content.

---

> ### Author Rebuttal · Authors · 2026-03-31
>
> **We sincerely thank the reviewer for the deep theoretical insights and constructive critique. We address your questions below.**
>
> ---
> ### **1. Degeneracy, Collapse, and Distinction with LN**
> **(Response to "Q1")**
>
> Please see **kmDA (P1)** for how LN’s degenerate Killing form makes it blind to center-dependent signals, leading to an “effectively linear” collapse. ReLN resolves this via the non-degenerate $\tilde{B}$, which drives all invariant operations: providing the gating scalar (ReLN-ReLU),  pooling (Max-Killing Pooling), and producing the final Readout.
>
> ---
> ### **2. Steerable Baselines, Irreps, and Tensor Structure**
> **(Response to "Q2 & Major Issue 1, 4")**
>
> Unlike field-based steerable networks (e.g., *Cohen et al. 2019*) that struggle with infinite-dimensional representations for non-compact groups or practical field-based implementations force truncation due to continuous spectra, ReLN enforces exact equivariance through closed-form algebra on finite carrier spaces.
>
> Furthermore, irrep decomposition shatters covariances (scale and orientation) into separate type-0/type-2 components, forcing complex combinatorial routing. ReLN’s full $\mathfrak{gl}(n)$ embedding preserves this holistic matrix geometry.
>
> **Left/Right Action (Section 4):** To clarify the tensor-shape roadmap: for a feature $x \in \mathbb{R}^{K \times C}$, the group acts on the geometric dimension $K$ (left), while weights act on channels $C$ (right). By matrix associativity, exact equivariance is guaranteed.
>
> ---
>
> ### **3. Lie-Theoretic Scope & Taxonomy**
> **(Response to "Major Issues 2,3")**
>
> The obstruction for $\mathfrak{gl}(n)$ is **non-semisimplicity**, not non-compactness; the text is revised. We also revise Fig 2 so $E(n)/SE(n)$ extends outside $GL(n)$. We also added the standard reference (*Kirillov, 2008*) for the decomposition $\mathfrak{g} = \mathfrak{z}(\mathfrak{g}) \oplus [\mathfrak{g},\mathfrak{g}]$ and the classical construction of non-degenerate completion by combining the Killing form on the semisimple summand with nondegenerate symmetric form on the center.
>
> ---
>
> ### **4. Scope of Claims, Error Bars, and Experimental Roadmap**
> **(Response to "Q3, Soundness, Originality")**
>
> We agree that the experiments should be easier to interpret relative to the paper’s claims. As detailed in kmDA (P2), the revised manuscript adds a Section 5 roadmap **mapping each benchmark to its input class, task symmetry, and intended claim**. We agree that the overlapping uncertainties in Tables 1 & 5 do not prove empirical superiority. As detailed in **kmDA (P2)**, they serve as **backward-compatibility** and **computational efficiency** checks.
>
> Drone State Estimation: We added 3-seed error estimates (ATE/RTE, m) in the revised manuscript. We report the strongest input configuration per model below, while extended results will be in the revision:
> * ResNet $(v,C)$: 208.1±2.4 / 107.8±1.5
> * VN $v$: 18.0±0.4 / 13.6±0.1
> * TFN $(v,\log C)$: 16.1±0.6 / 12.5±0.7
> * LN $(v,\log C)$: 15.6±0.1 / 11.9±0.1
> * **ReLN $(v,\log C)$**: **13.4±0.6 / 10.6±0.5**
>
> The non-overlapping variances confirm a statistically significant gain over LN.
>
> 3DGS Robustness:  We add the LN comparison in 3DGS to clarify the robustness :
> |Method|Acc.| Rotated Acc.|
> |:---|:---:|:---:|
> | Baseline| 93.39 | 18.28 |
> | Lie Neurons | 91.20 | 91.20 |
> | ReLN | 94.82 | 95.15 |
>
> The conclusion is more precise with added experimental validations: ReLN provides the clearest gains on center-sensitive tasks.
>
> Taken together with our responses to kmDA (W2) and cPMr (P1-3), these results clarify our intended claim: **ReLN is not a universal improvement over LN, but a principled extension whose advantages appear specifically in center-sensitive settings.** In the revised manuscript, we will revise the experimental presentation to better align the evidence with the paper’s claim.
>
> Presentation: Tab 3, Figs 4 & 6 are updated to include the LN configuration and fix the typo of `cov` to `log-cov`.
>
> ---
> ### **5. Clarity & Minor Issues**
> **(Response to "Presentation, Minor Issues")**
>
> As noted in kmDA (P3), Sec 4 is expanded to introduce the center, adjoint action, and $\tilde{B}$ before layer definitions. We will also revise the Introduction and Related Work to make the novelty stand out more clearly: the key contribution is a $GL(n)$ generalization in equivariant networks, which provides a unified Lie-algebra modeling perspective. Minor issues (L44R: corrected to "equivariant nonlinearities", Def 4.1 formatting including punctuation, and self-contained captions with clear tasks, symmetry, metrics) will all be fixed in the revision.
>
> ---
> References
>
> *Cohen et al., A General Theory of Equivariant CNNs on Homogeneous Spaces, NeurIPS (2019).*
>
> *Kirillov, An Introduction to Lie Groups and Lie Algebras, Cambridge Univ. Press (2008).*
>
> ---
> **We thank the reviewer again for the constructive questions. We hope these clarifications further highlight the robustness and rigor of our work.**

---

> > ### Author Rebuttal · Reviewer_jS9p · 2026-04-02
> >
> > I appreciate the author's thorough response to my questions and those of the other reviewers. Assuming that the theoretical discussion will be extended as described (e.g. kmDA P1), and the claims will be moderated and clarified in the context of this discussion, I have raised my initial score.

---

> > > ### Author Response · Authors · 2026-04-02
> > >
> > > **We sincerely thank Reviewer jS9p for the thorough evaluation, constructive feedback, and for deciding to raise the score.**
> > >
> > > ---
> > > We are glad that our detailed rebuttal and revision resolved your concerns. As you specifically noted, we will ensure that the theoretical discussion is thoroughly extended in the final manuscript (including **the claim of scope and novelty discussed in kmDA P1**). We will carefully moderate our claims within the context of this expanded discussion to ensure our contributions are accurately represented.
> > >
> > > ---
> > > We thank you again for your detailed feedback and positive evaluation. We believe these updates will significantly improve the presentation of our results and provide the necessary confidence in our findings.

---

### Official Review · Reviewer_cPMr · 2026-03-08

**Soundness:** 3
**Presentation:** 4
**Significance:** 3
**Originality:** 3
**Overall Recommendation:** 5
**Confidence:** 4

**Summary:**

This paper extends Lie Neurons from semisimple Lie algebras to reductive Lie algebras by replacing the degenerate Killing form with a non-degenerate Ad-invariant bilinear form, and then reusing a similar Lie-algebraic layer toolbox. The paper aims to provide a more general framework for equivariant learning under GL(n)-type linear symmetries.

**Compliance With Llm Reviewing Policy:**

Affirmed.

**Final Justification:**

The paper is technically sound and clearly presented. My main concerns were about the scope of the GL(n) claim, the motivation for the center completion, and whether the experiments sufficiently supported the broader framing. The rebuttal addressed most of these concerns by clarifying the distinction between native full GL(n)-equivariant tasks and covariance applications with reductive inputs, and by adding a more convincing discussion of the center and its practical role.

I still think the originality and empirical breadth are somewhat moderate, and the strongest evidence is in center-sensitive settings where the proposed framework is particularly well suited. I also note that, beyond the $gl(n)$ case considered in practice, higher-dimensional centers may introduce additional choices in the invariant inner product that could lead to different inductive biases. However, I view this more as a limitation of scope than as a flaw that changes my overall assessment here. Overall, the rebuttal improved my evaluation meaningfully, and I now find the contribution solid enough for acceptance, though my overall view remains between Accept and Weak Accept.

**Key Questions For Authors:**

1. Can the authors provide a task whose native symmetry is intrinsically reductive and center-dependent, rather than semisimple or an orthogonal/Lorentz case embedded into gl(n)?
2. Can the authors provide a more complete geometric justification for the claimed GL(n) applicability in covariance settings, beyond the orthogonal-frame case discussed in the limitations?
3. Can the authors better separate the claim of being a general framework from the more modest claim of being a useful extension of Lie Neurons?
4. I understand that Definition 4.1 gives an algebraically valid non-degenerate Ad-invariant completion of the Killing form. However, my main concern is about the choice of the center. The paper claims that one may choose any Ad-invariant inner product on $z(\mathfrak g)$, which makes this part of the construction seem somewhat arbitrary. In other words, while the degeneracy can certainly be removed this way, I do not understand why this particular completion should be viewed as a conceptually meaningful or well-motivated choice, rather than just one workable fix.
- 4.1) How sensitive is the method to this choice in principle?
- 4.2) Are different Ad-invariant inner products on $z(\mathfrak g)$ equivalent up to a harmless rescaling, or could they lead to genuinely different network behaviors?
- 4.3) Why should the specific choice used for $gl(n)$ be preferred beyond algebraic convenience?
- 4.4) It would also help if the authors could explicitly clarify where $\tilde B$ is used in the network, beyond the general statement that it serves as a basic contraction.

Overall, if the authors can satisfactorily resolve these questions and weaknesses, I would be willing to raise my score to Accept.

**Limitations:**

yes

**Strengths And Weaknesses:**

## Strengths:
1. The core mathematical idea is relatively simple and technically reasonable.
2. The paper is generally readable. I appreciate Table 7 for its clear summary. If space permits, moving a simplified version into the main text could make the paper’s motivation clearer.

## Weaknesses:
1. The main novelty still appears limited. Much of the architecture is inherited from Lie Neurons, and the main new ingredient is the replacement of the bilinear form on the reductive center.
2. The empirical evidence still does not fully support the broader GL(n) claim. On the semisimple benchmarks, the method is mostly comparable to Lie Neurons, with only limited gains.
3. Although the authors acknowledge a limitation regarding covariance, I consider this issue central rather than minor. The paper admits that congruence transformations coincide with adjoint conjugation only under orthogonal frame changes. Therefore, extending the same modeling story to general GL(n) coordinate changes still lacks sufficient geometric justification. In my view, this significantly weakens one of the paper’s main messages, namely that the proposed framework broadly captures general GL(n)-equivariant structure.
4. The top-tagging results are competitive, but they do not clearly demonstrate a strong advantage over the best baselines. This part reads more as evidence of compatibility than of clear empirical superiority.

---

> ### Author Rebuttal · Authors · 2026-03-31
>
> **We sincerely thank the reviewer for the careful reading and constructive feedback. Your questions target clear theoretical scope and empirical support, which we address below.**
>
> ---
> ### **1. Native $GL(n)$ Tasks**
> **(Response to "Q1, W2")**
>
> To demonstrate a native center-dependent task, we highlight two domains:
> - **Fluid mechanics:** The velocity-gradient $L = \nabla v \in \mathfrak{gl}(3)$ transforms by similarity. Its center $\mathrm{tr}(L)=\nabla \cdot v$ captures compressibility, which traceless models ignore.
> - **System Identification:** In learned state-space modeling, differing calibrations or latent representations naturally induce linear transformations ($x'=Tx$). For linear systems $\dot{x}=Ax+Bu$, this basis change induces similarity $A' = TAT^{-1}$, a native $GL(n)$ conjugation.
>
> Empirically, we added a $GL(3)$-equivariant **System ID benchmark**: recovering global dynamics $A$ from noisy local least-squares estimates $\tilde{A}\_i$ computed from trajectory windows of system $x_{k+1}=Ax_k+\epsilon_k$. We evaluate zero-shot generalization under unseen $GL(3)$ basis changes (GL and GL-Hard).
>
> We report Trace MSE (center recovery error) and Canonical MSE (intrinsic error $\|T^{-1}\hat{A}T - A_{\mathrm{gt}}\|_F^2$) over 3 seeds:
>
> |Method|Split|Trace MSE|Canonical MSE|
> |:---|:---|:---|:---|
> |Avg-LS|All | 0.0211 ± 0.0004 | 0.0044 ± 0.0002 |
> |MLP|ID|0.0145 ± 0.0005 |0.0041±0.0002|
> ||GL|0.0489±0.0064 | 0.1480 ± 0.0220 |
> ||GL-Hard|1.0285 ± 0.0270 | 57.4099 ± 7.3559 |
> |LN|All| 1.6850 ± 0.0500 | 0.0658 ± 0.0017 |
> |**ReLN**|**All** | **0.0133 ± 0.0005** | **0.0039 ± 0.0002** |
>
> MLP degrades severely under $GL(3)$ transforms. LN is stable but incurs large trace error by discarding the center. ReLN **preserves the center signal, ensuring robust generalization and improving slightly over the Avg-LS baseline.**
>
> ---
> ### **2. Geometric Scope: Covariance as Reductive Input**
> **(Response to "Q2, W3")**
>
> We agree this distinction needs more clarity. In covariance applications (3DGS, Drone Tracking), physical symmetry (sensor/viewpoint changes) is the orthogonal-frame case where congruence is conjugation ($A^{-1}=A^\top$). We do not claim arbitrary $GL(n)$ symmetry here, but a complementary regime with orthogonal task symmetry and reductive inputs.
>
> As detailed in our justification (**see kmDA, P2; TMWP, P3**), semisimple models fail here by discarding the center. ReLN provides an equivariant way to process covariance matrices by treating them as elements of $\mathfrak{gl}(n)$, preserving the full reductive structure, whereas LN fails to learn from the center of covariance.
>
> ---
> ### **3. Scope of the Claim**
> **(Response to "Q3, W1, W4")**
>
> We agree that the scope of our work should be made clearer. We refer to detailed responses on novelty and scope to kmDA (P1-2) and TMWP (P1). Architecturally, ReLN is a general reductive equivariant framework for both input/symmetry; empirically, its advantage over LN appears when the input or task involves a center. On semisimple benchmarks (e.g., top-tagging), ReLN matches LN, indicating compatibility. The core novelty is that our framework provides a unified modeling perspective: rather than tailoring architectures to specific symmetry, **a broad class of geometric problems can be expressed within a reductive feature representation**, unlocking $GL(n)$ equivariant tasks previously inaccessible to semisimple models.
>
> ### **To clarify our core thesis explicitly:** ###
> > *ReLN is a general $GL(n)$ adjoint-equivariant framework in its representation, while its clear empirical advantages emerge in center-sensitive settings. System ID test provides evidence for native full $\mathfrak{gl}(n)$ / $GL(n)$ equivariance, whereas covariance applications support the regime of reductive inputs.*
>
> ---
> ### **4. Choice of the Center Inner Product and Role of $\tilde{B}$**
> **(Response to "Q4.1-4")**
>
> For $\mathfrak{gl}(n)$, the center consists of scalar multiples of the identity. Consequently, Ad-invariant inner products on the center are equivalent up to a scalar rescale.
> - **Sensitivity:** In the revised paper, we will add the following sentence after Definition 4.1:
> > *For $\mathfrak{gl}(n)$, the center is 1D, different Ad-invariant inner products differ by a scalar rescale; in practice, we observe that scale does not affect the performance as the trainable weights absorb it during optimization.*
> - **Rationale:** Our chosen completion is preferred as it is the simple nondegenerate Ad-invariant extension of the Killing form: it agrees with the Killing form on the semisimple part while assigning a positive pairing to the center.
> - **Role of $\tilde{B}$:** $\tilde{B}$ drives all invariant operations: computing the scalar gate in ReLN-ReLU, the score in Max-Killing Pooling, and the final scalars in the Readout Layer.
>
> ---
> **We thank again for the detailed feedback. We believe these updates significantly improve the presentation of our results and reinforce the value of our contribution.**

---

> > ### Author Rebuttal · Reviewer_cPMr · 2026-04-02
> >
> > I appreciate the authors' effort in providing a thorough rebuttal. Most of my initial concerns have been resolved, but I would like to seek further clarification on the following four points:
> >
> > - The distinction between native full $GL(n)$ adjoint-equivariant tasks and covariance applications with orthogonal task symmetry plus reductive inputs is useful. However, it should be reflected explicitly in the article. The current presentation may overstate the scope supported by the experiments.
> >
> > - The discussion of the centre inner product seems convincing only for $\mathfrak{gl}(n)$. If all practical settings in the paper rely only on this case, this restriction should be stated clearly. If not, what happens for reductive Lie algebras with higher-dimensional centres?
> >
> > - The claim that the centre scale does not affect performance because trainable weights absorb it is currently informal. An ablation study would help verify the claim in practice.
> >
> > - The two added centre-dependent examples are helpful, but their task descriptions are still brief. It would help to explain them more concretely.
> >
> > ## Update
> >
> > The rebuttal has improved my evaluation. For the final version, I encourage the authors to make it more explicit that, beyond the $\mathfrak{gl}(n)$ case, higher-dimensional centres may allow non-equivalent choices of Ad-invariant inner products, which could lead to different inductive or geometric biases rather than amounting to a harmless rescaling.

---

> > > ### Author Response · Authors · 2026-04-06
> > >
> > > **We sincerely thank for the careful follow-up. Below, we clarify the remaining points.**
> > >
> > > ---
> > > ### **1. Scope of the Experimental Claims**
> > > **(Follow-up P1)**
> > >
> > > As noted in kmDA (P2) and TMWP (P3), the revision will clearly separate:
> > > - **native full \(GL(n)\) adjoint-equivariant tasks**, where both the input representation and task symmetry require the full reductive setting, and
> > > - **covariance applications with orthogonal task symmetry**, where the physical symmetry is typically orthogonal, the input still contains a nontrivial center that semisimple architectures discard.
> > >
> > > We revise the Introduction and Sec. 5 to include a benchmark roadmap mapping experiments to (i) the input class, (ii) the task-level symmetry, and (iii) the claim it supports. This will make explicit that System Id provides primary evidence for full $GL(n)$ equivariance, while covariance-based tasks support the reductive matrix-valued inputs. A modified claim with this roadmap will clarify the scope of our work.
> > >
> > >
> > > ### **2. Higher-Dimensional Centres and the Theoretical Scope**
> > > **(Follow-up P2)**
> > >
> > > At the theory level, our construction is not restricted to $\mathfrak{gl}(n)$. By Ado’s theorem, any finite-dimensional reductive Lie algebra can be realized as a Lie subalgebra of $\mathfrak{gl}(N)$ for some $N$, allowing ReLNs to be implemented on matrix representatives. That said, our work on reductive Lie algebras does not rely only on this embedding viewpoint. The key point is the reductive decomposition $\mathfrak g = z(\mathfrak g)\oplus[\mathfrak g,\mathfrak g],$ from which we construct a non-degenerate invariant bilinear form by combining an Ad-invariant positive-definite inner product on the center with the Killing form on the semisimple ideal. This remains valid even when the center is higher-dimensional. In the revised paper, we will clarify this general scope and distinguish it from our focus on the matrix setting of $\mathfrak{gl}(n)$
> > >
> > > ### **3. Centre-Scale Sensitivity**
> > > **(Follow-up P3)**
> > >
> > > To verify that the center scale is absorbed during training, we conducted an ablation study on the System ID task. We varied the scale factor $\alpha > 0$ applied to the center's inner product. For $\mathfrak{gl}(3)$, this corresponds to the trace formulation:  $\tilde B_{\alpha}(X,Y) = B(X, Y) + \alpha \langle X_{c}, Y_{c} \rangle = 6\mathrm{Tr}(XY) + (\alpha - 2)\mathrm{Tr}(X)\mathrm{Tr}(Y)$.
> > >
> > > |Center Scale ($\alpha$)|Trace MSE|Canonical MSE|
> > > |:---|:---|:---|
> > > |0.1|0.0135 ± 0.0002|0.0040 ± 0.0002|
> > > |0.5|0.0127 ± 0.0006|0.0038 ± 0.0002|
> > > |1.0 |0.0133 ± 0.0005|0.0039 ± 0.0002|
> > > |2.0|0.0130 ± 0.0004|0.0039 ± 0.0002|
> > > |10.0| 0.0140 ± 0.0004|0.0041 ± 0.0002|
> > >
> > > Across two orders of magnitude in $\alpha$, the final metrics remain unchanged. This supports that the scale at the center is absorbed during optimization. We will include this ablation in the appendix.
> > >
> > > ### **4. Details of Centre-Dependent Examples**
> > > **(Response to Follow-up P4)**
> > >
> > > In the revised manuscript, we will clarify the learning task, transformation law, and role of the center in both examples.
> > >
> > > For **Fluid Mechanics (Compressible Flow Modeling)**, we explain that the relevant object is the **velocity-gradient tensor** $L=\nabla v \in \mathfrak{gl}(3)$, which transforms by similarity under a linear change of frame. In this decomposition, the traceless part captures shear and rotational structure, whereas the center corresponds to the isotropic component $\frac{\mathrm{tr}(L)}{3}I$, i.e., the divergence $\nabla \cdot v$, which measures local volumetric expansion. This is a learning setting in which the center is physically meaningful: a semisimple formulation removes this component and therefore cannot represent compressibility, whereas the reductive formulation preserves both the deviatoric and isotropic parts within a single equivariant representation.
> > >
> > > For **System Identification (Latent Dynamics Learning)**, we will explain that we consider discrete-time linear systems
> > > $x_{k+1}=A_{\mathrm{gt}}x_k+\epsilon_k,$ with input features $\tilde A_i \in \mathfrak{gl}(n)$ given by noisy local transition matrices estimated from short trajectory windows. The task is to learn these local estimates and recover the true global dynamics operator $A_{\mathrm{gt}}$. Under a change of latent basis or calibration, the dynamics transforms by similarity, making this a native $GL(n)$ adjoint-equivariant task. The center corresponds to the trace component, which carries global expansion information. We clarify that this benchmark is not only a symmetry check, but a recovery problem in which discarding the center removes part of the target operator. We connect this to the reported metrics: Trace MSE measures recovery of the center-sensitive part, while Canonical MSE measures intrinsic recovery modulo basis change.
> > >
> > > ---
> > > **We thank you again for your detailed feedback. We believe these updates will improve the clarify the scope of our claims, deepen the discussion, and provide stronger support for our findings.**

---

### Official Review · Reviewer_TMWP · 2026-03-10

**Soundness:** 3
**Presentation:** 3
**Significance:** 2
**Originality:** 2
**Overall Recommendation:** 4
**Confidence:** 3

**Summary:**

The authors introduce a new architecture called Reductive Lie Neurons (ReLNs), which is $GL(n)$-equivariant by construction. Since the Lie algebra $\mathfrak{gl}(n)$ is reductive but not semisimple, its standard Killing form is degenerate. This structural limitation restricted previous Ad-equivariant frameworks to semisimple algebras. To address this, the authors introduce a modified, non-degenerate bilinear form, $\tilde{B}$, which is Ad-invariant by construction. This form orthogonally decomposes the algebra, applying the standard Killing form $B$ to the semisimple ideal (the traceless component) and a standard inner product to the center. This resolves the degeneracy issues on reductive algebras, enabling stable invariant gating and exactly equivariant nonlinearities. The authors provide empirical evidence of ReLNs' efficacy across different domains, including algebraic benchmarks on $\mathfrak{sl}(3)$ and $\mathfrak{sp}(4)$, uncertainty-aware drone state estimation, 3D Gaussian Splatting representation learning, and Hamiltonian dynamics simulations.

**Compliance With Llm Reviewing Policy:**

Affirmed.

**Final Justification:**

I appreciate the authors' response. Though I feel that the paper will be stronger if there are more theoretical results regarding convergence or asymptotic sample complexity, the current form also meets my standard of acceptance. Therefore, I tend to accept this paper.

**Key Questions For Authors:**

- Could the authors clarify point (3) in weaknesses? If the practical handling of data restricts the physical interpretation entirely to $SO(n)$, which is already semisimple for n >= 3, what is the main benefit of using this heavier reductive machinery rather than existing semisimple architectures?
- Given the strong empirical performance of ReLNs across the demonstrated benchmarks, can the authors provide any formal theoretical or asymptotic guarantees (e.g., regarding sample complexity, convergence, or representational capacity) to mathematically support these empirical results?

**Limitations:**

yes

**Strengths And Weaknesses:**

**Strengths**
- The paper is generally well-written. Although the theoretical preliminaries are quite dense, I liked the figures and table summaries the authors provide to aid the understanding.
- The primary strength of this work lies in its experimental section. The authors convincingly demonstrate the performance of Reductive Lie Neurons (ReLNs) across a diverse set of domains (e.g., algebraic benchmarks, drone state estimation, and 3D vision).
- The construction of the architecture to achieve exact equivariance on $\mathfrak{gl}(n)$ is an elegant and practical idea that can be beneficial for downstream tasks requiring broad linear symmetries.

**Weaknesses**
- The core mathematical construction relies heavily on the framework established in [Lin et al., 2024a] for semisimple algebras (bilinear form B). The authors extend this to the reductive Lie algebra $\mathfrak{gl}(n)$ by adding a standard inner product term on the center (thus obtaining $\tilde{B}$). While mathematically sound, the methodology (in my understanding) seems to be a direct structural extension of prior work.
- Fundamental theoretical claims in the paper are limited. Hence, the paper leans heavily on its empirical results to prove its value. While the experiments are well-designed to test the algebraic properties, they rely almost entirely on small-scale, synthetic, or highly controlled simulated datasets. The architecture's robustness on complex, noisy, real-world data seems to be unproven.
- While the paper's central premise is achieving broad $GL(n)$ equivariance, the way it actually handles geometric data like covariances relies on equating $A \Sigma A^T$ with $A \Sigma A^{-1}$. This only works for orthogonal matrices. This is a significant limitation because it restricts the model's physical interpretation entirely to $SO(n)$ rotations. Furthermore, since $SO(n)$ is already semisimple for $n \geq 3$, why introduce this heavy reductive machinery at all? This limits the practical scope of the work and undermines the broader $GL(n)$ claims.

---

> ### Author Rebuttal · Authors · 2026-03-30
>
> **We thank the reviewer for the constructive and insightful feedback. We appreciate the comments on the paper’s scope, empirical validation, and level of claim.**
>
> ---
> ### **1. Structural Continuity, Novelty, and Practical Necessity**
> **(Response to "Weakness 1")**
> > The core mathematical construction relies... While mathematically sound...
>
> We agree that ReLN preserves the architectural backbone of VN and LN, and some implementation-level resemblance is therefore expected. For the architectural lineage and the precise point at which the novelty enters, please see our response to kmDA (P1). Briefly, the contribution is the resolution of the obstruction that prevents semisimple LN from extending to reductive Lie algebras $\mathfrak{gl}(n)$. As the Killing form is degenerate at the center, LN is blind to center signals. ReLN resolves this with the non-degenerate completion $\tilde{B}$, restoring center-sensitive contractions on the full reductive feature space.
>
> The empirical necessity of this extension is summarized in kmDA (P2) and further supported by the added System ID results in cPMr (P1). In the revised manuscript, this is made explicit through the benchmark rationale in Section 5 and the motivating example in Section 4 (described in kmDA, P3). Together, these revisions clarify that ReLN is a general completion for a reductive Lie group/algebra.
>
> ---
> ### **2. Controlled Algebraic Validation vs. Practical Application Evidence**
> **(Response to "Weakness 2")**
> > Fundamental theoretical claims... The architecture's robustness on complex ...
>
> We agree on the importance of large-scale, real-world validation. The goal of this work, however, is to establish a rigorous foundation for reductive equivariance and demonstrate its applicability across diverse domains (algebraic learning, physics-inspired dynamics, and 3D computer vision), rather than optimize for a single deployment setting.
>
> The experimental section goes beyond controlled algebraic validation. The chosen benchmarks are standard in the literature and cover distinct regimes in which reductive structure matters. Beyond algebraic tests, the paper includes 3D GS object vision on ShapeNet/ModelNet, uncertainty-aware drone motion estimation, and the EMLP double-pendulum benchmark.
>
> To clarify the role of each experiment, the revised manuscript includes the benchmark-roadmap paragraph referenced in kmDA (P2), which maps covariance/3DGS to center-sensitive inputs, EMLP to $GL(n)$ efficiency, and semisimple benchmarks to backward compatibility. Larger-scale real-world validation remains an important direction for future work.
>
> ---
> ### **3. Why Reductive Machinery Is Needed**
> **(Response to" Weakness 3, Key Questions 1")**
> > While the paper's central premise is achieving broad $GL(n)$...
>
> The key distinction is in **task-level symmetry** and **input-level generalization**. We agree that covariance-type data are typically interpreted under $A\Sigma A^T$, and in practical settings this reduces to $SO(n)$ rather than full $GL(n)$ symmetry. This does not make semisimple architectures sufficient.
>
> Semisimple architectures such as LN operate on traceless features and are blind to the center. Applied to covariance matrices, they remove the trace component and therefore lose information carried by scale or uncertainty magnitude. ReLN is therefore needed not only for full $GL(n)$ tasks, but also for reductive matrix-valued inputs even when the symmetry is orthogonal.
>
> This distinction is made explicit in the revised manuscript through the benchmark rationale detailed in our response to Reviewer kmDA (P2) and the added System ID clarification in our response to Reviewer cPMr (P1). System Identification covers the regime in which both the input and the symmetry require the full reductive framework, while covariance-based tasks cover the complementary regime in which the symmetry may be restricted to $SO(n)$ but the input still contains a nontrivial center that semisimple methods cannot preserve. In short, the reductive machinery is needed both for genuinely full $GL(n)$ tasks and for center-sensitive matrix inputs that semisimple architectures discard.
>
> ---
> ### **4. Formal Theoretical and Asymptotic Guarantees**
> **(Response to "Key Questions 2")**
> > Given the strong empirical performance of...
>
> We thank the reviewer for highlighting these important problems. Currently, we do not have formal theoretical results regarding convergence or asymptotic sample complexity. While ReLN benefits from reduced sample complexity by restricting the hypothesis class to $GL(n)$-equivariant functions, formally proving these bounds is beyond the current scope. We will explicitly discuss these limitations and outline the mathematical derivation of these guarantees as a key direction for future work in the revised paper.
>
> ---
> **We thank the reviewer again for the constructive feedback and hope these updates clarify our contribution and address the concerns effectively.**

---

> > ### Author Rebuttal · Reviewer_TMWP · 2026-03-31
> >
> > I appreciate the authors' response. I decide to raise my score since the authors resolve some of my concerns. However, I feel that the paper will be stronger if there are more theoretical results regarding convergence or asymptotic sample complexity.

---

> > > ### Author Response · Authors · 2026-04-06
> > >
> > > **We sincerely thank Reviewer TMWP for the positive evaluation and for deciding to raise the score.**
> > >
> > > ---
> > > We agree that establishing formal theoretical guarantees, such as convergence or asymptotic sample complexity, would further strengthen the foundation of ReLNs. While deriving asymptotic bounds is beyond the empirical scope of the current work, your feedback highlights an essential direction. As you suggested, we will explicitly include a discussion/direction in the revised manuscript outlining these theoretical explorations for future work. Universal Approximation Property (UAP) is a critical open question for the proposed method. The next theoretical step is to investigate whether a network composed of ReLN primitives can approximate continuous function within the class of $GL(n)$ adjoint-equivariant functions.
> > >
> > > We will ensure that all the clarifications, additional discussions, and new results provided during this rebuttal phase are thoroughly incorporated into the updated manuscript.
> > >
> > > ---
> > > **We thank you again for your constructive guidance. We believe these theoretical discussions, alongside the empirical updates, will significantly improve the completeness of our work.**

---

### Official Review · Reviewer_kmDA · 2026-03-13

**Soundness:** 4
**Presentation:** 3
**Significance:** 3
**Originality:** 2
**Overall Recommendation:** 4
**Confidence:** 3

**Summary:**

The paper,  “Equivariant Neural Networks for General Linear Symmetries on Lie Algebras” proposes an architecture for exact adjoint-equivariant learning on reductive Lie algebras, with a particular emphasis on  $\mathfrak{gl}(n)$  and, consequently, general linear symmetries. The central technical idea is to replace the degenerate Killing form on reductive algebras with a non-degenerate Ad-invariant bilinear form, which is then used to define invariant contractions, gating, pooling, and nonlinear layers. Empirically, the method is evaluated on semisimple algebra benchmarks, drone state estimation with velocity and covariance inputs, 3D Gaussian splat representation learning, a double-pendulum benchmark against EMLP, and Lorentz-equivariant top-tagging in the appendix.

**Compliance With Llm Reviewing Policy:**

Affirmed.

**Key Questions For Authors:**

See weaknesses.

**Limitations:**

Yes.

**Strengths And Weaknesses:**

**Strengths**
-   The paper is clearly written overall, and Figure 2 offers a useful high-level taxonomy.
-   The handling of Killing form degeneracy is simple, elegant, and technically well motivated.
-   The experimental setup is solid, and the results on drone state estimation are especially strong.

**Weaknesses**

-   **Limited architectural novelty compared with LieNeurons [1]:**  While the proposed treatment of degeneracy is elegant, much of the architecture appears to be a direct extension of LieNeurons [1]. This reduces the perceived novelty of the work. In particular, ReLN-ReLU and ReLN-Bracket read more as natural adaptations than as genuinely new architectural components.

-   **Presentation may be inaccessible to part of the target audience:**  The ideas and presentation in section 4 is mathematically dense and may be difficult to follow for readers without prior familiarity with reductive Lie algebras. Although Appendix A provides some supporting background, essential concepts such as the center of a Lie algebra and the meaning of  $[\mathfrak{g}, \mathfrak{g}]$ are not introduced explicitly. A more gradual and self-contained exposition would improve accessibility and broaden the paper’s impact within the machine learning community.

---

> ### Author Rebuttal · Authors · 2026-03-30
>
> **We thank the reviewer for the constructive comments and the recognition of our framework’s practical impact.**
>
> ---
> > **Limited architectural novelty compared with LieNeurons**...
> ### **1. Novelty and the "Effectively Linear" Collapse**
> **(Response to “Weakness 1”)**
>
> We appreciate the reviewer’s perspective that architectural resemblance affects the *perceived* novelty. However, this similarity is expected as ReLN is a mathematical generalization of LN: as LN extends Vector Neurons from the isomorphic $\mathfrak{so}(3) \simeq \mathbb{R}^3 $ setting, ReLN extends LN from semisimple to reductive Lie algebras. The required representation-theoretic extension is nontrivial, while the streamlined geometric backbone is deliberately preserved.
>
> The primary novelty is resolving the obstruction to extending semisimple frameworks to $\mathfrak{gl}(n)$: the degeneracy of the Killing form on the center. In particular, for any center element $Z = aI$, one has $B_{\mathfrak{gl}(n)}(Z, Y) = 0$ for all $Y \in \mathfrak{gl}(n)$. In LN, this degeneracy makes the network structurally blind to center-dependent signals (e.g., scale, trace). Concretely, if $X_1 = X_0 + Z_1$ and $X_2 = X_0 + Z_2$ (with $X_0 \in \mathfrak{sl}(n)$ and $Z_1 \neq Z_2 \in \mathfrak{z}$), LN cannot distinguish them. The invariant scalar used for non-linear gating (ReLU) becomes exactly $0$ on this subspace, so the layer collapses to effectively linear channel mixing ($x \mapsto xW$). ReLN resolves this via a nondegenerate completion $\tilde{B}$.
>
> Beyond resolving this collapse, the extension to $GL(n)$ provides a unified modeling perspective: rather than tailoring architectures to specific symmetry groups, a broad class of geometric problems can be expressed within a common $GL(n)$ representation. We will revise the Introduction and Related Work to make this perspective explicit and to distinguish the contribution more clearly from the semisimple LN setting.
>
> ---
> ### **2. Two Axes of Generalization & Supporting Experimental Roadmap**
>
> To prove this architectural necessity and facilitate interpretation of our theoretical claims, we discuss the experimental roadmap and the distinction between **input-level reductive structure** and **task-level symmetry**. In the revised manuscript, we will add the following paragraph at the start of Section 5 that adds a short rationale mapping each benchmark to (i) the input class, (ii) the task-level symmetry, and (iii) the claim it is intended to test:
>
> > *Each benchmark is designed to test a distinct aspect of the proposed architecture. System Identification evaluates symmetry-level generalization in the full $GL(n)$ regime, where both the input space and the task symmetry require the full reductive setting. This is the regime in which semisimple LN is insufficient. Benchmarks such as 3DGS and drone tracking instead evaluate input-level generalization in settings where the input features contain a nontrivial center, for example, scale or uncertainty encoded in covariance-like features. In such cases, semisimple LN removes the center by construction, whereas ReLN preserves it, making the reductive formulation necessary even when the task symmetry itself may be more restricted. We further evaluate computational efficiency relative to fully general $GL(n)$-equivariant numerical approaches based on constraint solving, such as EMLP. Finally, we include semisimple benchmarks in which the center is absent, showing that ReLN matches LN in the purely semisimple regime and therefore remains backward compatible when the additional reductive generality is not needed.*
>
> **To support the first claim empirically, we include detailed System ID experiments in response to Reviewer cPMr, Point 1**. These results show that our model captures full $GL(n)$ symmetry and learns effectively from $\mathfrak{gl(n)}$ inputs, whereas the compared methods fail on this task.
>
> ---
> ### **3. Presentation Clarity and Gradual Exposition**
> **(Response to “Weakness 2”)**
> > **Presentation may be inaccessible to part of the target audience:**...
>
> We agree with this comment and will make the exposition more self-contained in the revised manuscript.
>
> At the start of Section 4, we will add a roadmap paragraph, and before the layer definitions (Section 4.2), we will reorganize the presentation to introduce the center, reductive decomposition, adjoint action, and the completed bilinear form. We will also include a matrix example in which two inputs share the same semisimple part but differ in their center component, illustrating why LN becomes blind to center-dependent signals before we present the full formalism. Together, these changes make the presentation more accessible and provide geometric intuition for why the reductive completion is needed while preserving mathematical precision.
>
> ---
> **We thank the reviewer again for the generous feedback and helpful suggestions. We hope these clarifications address your concerns and clarify our contribution.**

---

> > ### Author Rebuttal · Reviewer_kmDA · 2026-04-01
> >
> > I thank the authors for the detailed rebuttal. Provided that the authors include the rebuttal in their updated manuscript, I retain my assessment of the paper and recommend it for acceptance with my original score.

---

> > > ### Author Response · Authors · 2026-04-01
> > >
> > > **We sincerely thank Reviewer kmDA for the positive evaluation and for recommending our paper for acceptance.**
> > >
> > > ---
> > > We are glad that our detailed rebuttal resolved all your concerns. As you requested, we will ensure that all the clarifications, additional discussions, and new results provided during this rebuttal phase are carefully and thoroughly incorporated into the updated manuscript.
> > >
> > > ---
> > > We thank you again for your detailed feedback. We believe these updates will significantly improve the presentation of our results and provide the necessary confidence in our findings.

---

### Decision · Program_Chairs · 2026-04-30

**Decision:**

Accept (regular)

**Comment:**

This paper derives equivariant architectures for the general linear group, and shows the derived method has practical advantages for tasks such as uncertainty-aware drone state estimation and learning from 3D Gaussian-splat representations. After discussion all reviewers recommend acceptance, as do I. In the camera ready version, I request that the author keep their promise to make the paper more accessible to the general ML audience. In particular, I would emphasize explanation of when and where the type of equivariance discussed in this paper is needed in applications.